# Cripto shapes macrophage plasticity and restricts EndMT in injured and diseased skeletal muscle

Francescopaolo Iavarone[1], Ombretta Guardiola[1], Alessandra Scagliola[2], Gennaro Andolfi[1], Federica Esposito[1], Antonio Serrano[3], Eusebio Perdiguero[3], Silvia Brunelli[2], Pura Muñoz-Cánoves[3,4,5] & Gabriella Minchiotti[1,*] (iD)

## Abstract

Macrophages are characterized by a high plasticity in response to changes in tissue microenvironment, which allows them to acquire different phenotypes and to exert essential functions in complex processes, such as tissue regeneration. Here, we report that the membrane protein Cripto plays a key role in shaping macrophage plasticity in skeletal muscle during regeneration and disease. Conditional deletion of Cripto in the myeloid lineage (Cripto[My-LOF]) perturbs MP plasticity in acutely injured muscle and in mouse models of Duchenne muscular dystrophy (mdx). Specifically, Cripto[My-LOF] macrophages infiltrate the muscle, but fail to properly expand as anti-inflammatory CD206[+] macrophages, which is due, at least in part, to aberrant activation of TGFβ/Smad signaling. This reduction in macrophage plasticity disturbs vascular remodeling by increasing Endothelial-to-Mesenchymal Transition (EndMT), reduces muscle regenerative potential, and leads to an exacerbation of the dystrophic phenotype. Thus, in muscle-infiltrating macrophages, Cripto is required to promote the expansion of the CD206[+] anti-inflammatory macrophage type and to restrict the EndMT process, providing a direct functional link between this macrophage population and endothelial cells.

**Keywords** Cripto; Duchenne muscular dystrophy; Endothelial-to-Mesenchymal Transition; macrophage plasticity; skeletal muscle regeneration

**Subject Categories** Immunology; Molecular Biology of Disease; Signal Transduction

## Introduction

Skeletal muscle regeneration relies on highly coordinated sequential events, which involve a complex network of interaction between tissue-resident and recruited cells, including muscle stem cells, inflammatory cells, endothelial-derived progenitors, and fibro/adipogenic progenitors (FAPs) [1]. Progressive impairment of the interplay between the inflammatory cells, mainly the infiltrated macrophages, and the different muscle cellular components is emerging as a key event in switching regeneration from compensatory to pathogenic regeneration [2]. Macrophages exhibit substantial plasticity and can quickly change their phenotype in response to different stimuli in the microenvironment. This unique feature makes them serve different functions during the process of skeletal muscle regeneration, including resolution of the pro-inflammatory phase and transition to the regeneration phase. Soon after an acute injury, invading macrophages adopt a pro-inflammatory phenotype, phagocyte necrotic tissue, and secrete growth factors, promoting an environment that favors satellite cell activation and proliferation [3–5]. Subsequently, a second wave of macrophages with anti-inflammatory phenotype suppresses the process of inflammation and supports restorative function. These anti-inflammatory macrophages arise from resident macrophages and/or circulating pro-inflammatory macrophages that switch into an anti-inflammatory phenotype [6]. Besides supporting myoblast differentiation, anti-inflammatory macrophages produce angiogenic factors promoting vascular remodeling, which is necessary for a proper regenerative process [7]. Furthermore, macrophages promote the differentiation of endothelial-derived progenitors by inhibiting a process known as the Endothelial-to-Mesenchymal Transition (EndMT) process, in which endothelial cells acquire myofibroblastic traits while losing endothelial-specific gene expression [8–10]. While in acute muscle injury, macrophages adopt well-coordinated sequential waves of pro- and anti-inflammatory phenotypes, this balance is skewed in chronic muscle diseases like muscular dystrophies, where muscles fail to regenerate and the tissue is progressively substituted by adipocytes and collagen fibers [11,12]. Emerging evidence indicates that different macrophage populations with mixed phenotypes coexist in dystrophic muscles, but their precise function is only recently starting to be

1   Stem Cell Fate Laboratory, CNR, Institute of Genetics and Biophysics "A. Buzzati-Traverso", Naples, Italy
2   School of Medicine and Surgery, University of Milano-Bicocca, Monza, Italy
3   Cell Biology Group, Department of Experimental and Health Sciences, Pompeu Fabra University (UPF), CIBER on Neurodegenerative Diseases (CIBERNED), Barcelona, Spain
4   Institució Catalana de Recerca i Estudis Avançats (ICREA), Barcelona, Spain
5   Centro Nacional de Investigaciones Cardiovasculares (CNIC), Madrid, Spain
    *Corresponding author. Tel: +39 0816 132357; E-mail: gabriella.minchiotti@igb.cnr.it

characterized [13]. Research efforts are currently focused on understanding how macrophage plasticity is controlled and how macrophages crosstalk with other muscle populations in injured and dystrophic muscles. Recent findings from our laboratory and others place the extracellular membrane protein Cripto within this complex regulatory network. Cripto is a glycosylphosphatidylinositol (GPI)-anchored protein and acts as coreceptor for different members of the TGF-β superfamily [14]. Of note, depending on the context it also exists as a soluble protein either secreted or shed from the membrane [15,16]. *Cripto* is a developmental gene known to regulate early embryonic development [17,18] and it is usually not expressed in normal adult tissues including resting skeletal muscles. However, Cripto becomes rapidly and transiently re-expressed upon acute injury both in activated/proliferating satellite cells and in a subpopulation of infiltrating macrophages [19]. Emerging evidence indicates that Cripto is a key regulator of the myogenic program during skeletal muscle regeneration [16,19,20]. However, no studies have been reported so far on the inflammatory cell/macrophage-specific role of Cripto in skeletal muscle regeneration. Here, we show *in vivo* evidence that Cripto is preferentially expressed by anti-inflammatory macrophages and is a key regulator of macrophage plasticity in injured and in dystrophic skeletal muscles. Moreover, we suggest that Cripto mediates the crosstalk between macrophages and endothelial cells promoting vascular remodeling, at least in part, by restricting TGF-β-induced EndMT and preventing excessive fibrosis in dystrophic muscles.

# Results

## Expression profile of cell-surface Cripto in subpopulations of macrophages during acute muscle injury

The timely attraction of macrophages (MPs), the ordered transition between the pro- and anti-inflammatory phenotypes, and the precise termination of their activity are prerequisites for a successful regeneration process [4]. Cripto is expressed in the MPs that infiltrate the injured muscle [19]; yet, the dynamics of Cripto expression in the pro- and anti-inflammatory MPs was not investigated so far. To address this issue directly, hind limb muscles of wild-type mice were injected with cardiotoxin (CTX) and the expression of Cripto in the different inflammatory cell populations was analyzed at days 2, 3, and 5 after injury (Fig 1A). To this end, injured muscles were dissociated by enzymatic digestion and the bulk cells were stained for Cripto and the MP markers CD11b, F4/80, and Ly6C. We first assessed the expression of Cripto in the $CD11b^+$ immune cells at the different time points after injury; the number of $CD11b^+$ that expressed Cripto progressively increased from days 2 to 5 up to 40% ($6.5 \pm 0.7\%$ at day 2 vs. $40.7 \pm 1.3\%$ at day 5; $n = 5$; $P = 1.14E-08$; Figs 1B and EV1A), consistent with previous data of Cripto expression in the infiltrating MPs [19]. We then examined the distribution of pro-inflammatory ($F4/80^+/Ly6C^{High}$) and anti-inflammatory ($F4/80^+/Ly6C^{Low}$) MPs in the $CD11b^+$ cells throughout the time course (Fig 1C). As expected, while the $F4/80^+/Ly6C^{Low}$ MPs (blue) and $F4/80^+/Ly6C^{High}$ MPs (red) were almost equally distributed in the $CD11b^+$ cells at day 2, from day 3 onward a dramatic increase in $F4/80^+/Ly6C^{Low}$ MPs was observed, at the expense of the $F4/80^+/Ly6C^{High}$ MPs (Fig 1C and D). Finally, to

directly examine the nature of the $Cripto^+$ immune cells, we evaluated the distribution of $F4/80^+/Ly6C^{High}$ and $F4/80^+/Ly6C^{Low}$ MPs in the $CD11b^+/Cripto^+$ population. Of note, the $F4/80^+/Ly6C^{Low}$ MPs were the vast majority at all the time points analyzed (Fig 1C and D).

The gradual increase in Cripto expression levels in the $CD11b^+/F4/80^+/Ly6C^{Low}$ MP population was detected by qRT–PCR (Fig EV1B). Collectively, these results indicate that Cripto progressively increased in the immune cells that infiltrate the injured muscle and predominantly accumulated at the surface of anti-inflammatory MPs, concomitantly with their accumulation in the regenerating tissue.

## Cripto controls MP plasticity

To investigate the physiological role of Cripto in the infiltrating MPs without interfering with its activity in the myogenic compartment [19], we generated myeloid-specific Cripto knockout (KO) mice. *Tg:Cripto^{fl/fl}* conditional mice [21] were crossed with the lysozyme M (LysM) Cre line (*Tg:LysMCre*) in which the Cre recombinase is expressed in the myeloid lineage [22]. Homozygous conditional *Tg:LysMCre::Cripto^{−/−}* mice (Cripto^{My-LOF}) were born in Mendelian ratio and developed normally. Acute injury was thus induced in tibialis anterior (TA) muscles of Cripto^{My-LOF} and Control mice by CTX injection and injured muscles were analyzed at days 2 and 5 after injury. Deletion of *Cripto* in the infiltrated MPs was verified by PCR (Appendix Fig S1A and B). Immunofluorescence analysis of injured muscle sections stained with F4/80 showed no significant difference in the $F4/80^+$ area between Cripto^{My-LOF} and Control mice at both days 2 and 5 after injury (Fig 2A and B), suggesting that *Cripto* genetic ablation did not affect MP accumulation. Given the expression of Cripto in the anti-inflammatory MPs, we analyzed this MP population at days 2 and 5 after injury by staining muscle sections for the anti-inflammatory marker CD206 (Fig 2C). While the frequency distribution of $CD206^+$ MPs was comparable in Cripto^{My-LOF} and Control muscles at day 2 (Fig 2C top panels and D), a significant decrease in $CD206^+$ MPs was observed in Cripto^{My-LOF} at day 5 (Fig 2C bottom panels and E). In order to specifically identify and characterize Cripto KO MPs, we combined conditional lineage-tracing and genetic ablation of *Cripto*. Cripto^{My-LOF} mice were thus crossed with the *Tg:R26^{mTmG}* transgenic line [23] to obtain the *Tg:LysMCre::R26^{mTmG}::Cripto^{fl/fl}* (GFP-Cripto^{My-LOF}; Fig EV2A and B) and the *Tg:LysMCre::R26^{mTmG}* (GFP-Control) as control. We first assessed the efficiency of LysMCre-mediated *Cripto* deletion by genomic qPCR analysis on Florescence-Activated Cell Sorting (FACS)-sorted $F4/80^+$ MPs (Cripto^{My-LOF}) or $GFP^+$ monocytes/MPs (GFP-Cripto^{My-LOF}). Genomic DNA from heterozygous Cripto KO (Cripto^{+/−}) mice was used as reference for 50% of gene deletion. The efficiency of *Cripto* deletion was ~ 50% in the overall MP population, while it increased up to ~ 80% in the $GFP^+$ population (Fig EV2C). Accordingly, ~ 50% of the MP population expressed the reporter GFP both in GFP-Cripto^{My-LOF} and GFP-Control mice (Fig EV2D). Finally, we measured Cripto protein by ELISA and found that it was significantly reduced in Cripto KO MPs compared to control (Fig EV2E).

Given that the vast majority (> 90%) of $GFP^+$ cells expressed F4/80 in both GFP-Cripto^{My-LOF} and GFP-Control muscles (Fig 3A), we quantified the percentage of $GFP^+$ cells/MPs at both

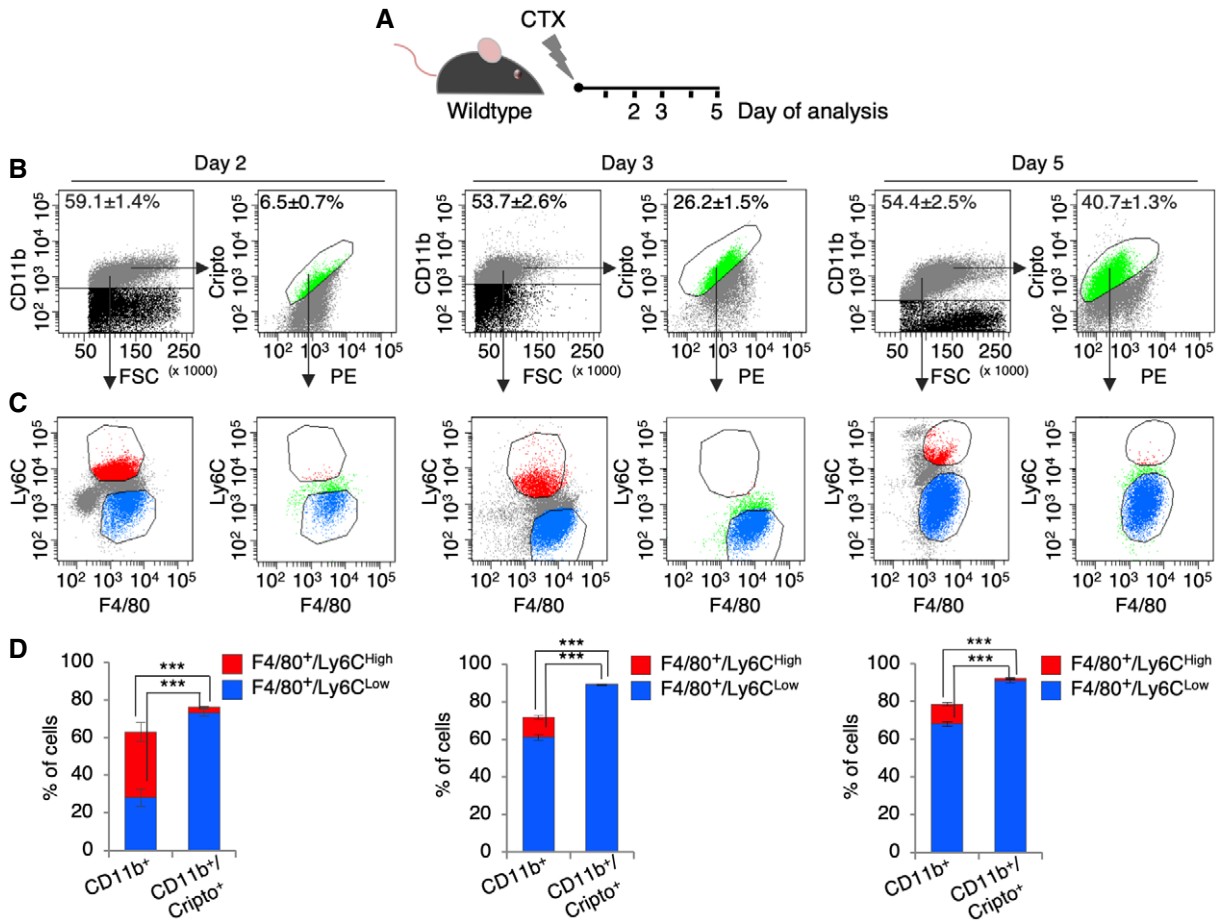

**Figure 1.  Time-course expression of surface Cripto in pro- and anti-inflammatory macrophages in acute skeletal muscle injury.**

A   Experimental scheme of Cripto expression analysis in macrophage (MP) subpopulations from wild-type skeletal muscles at different time points after cardiotoxin (CTX) injection.

B   Representative flow cytometry dot plots of CD11b[+] and CD11b[+]/Cripto[+] cells at days 2, 3, and 5 after injury. PE (Phycoerythrin/free channel). Data are mean ± SEM ($n$ = 5 biological replicates; $P \leq 0.00008$, Student's $t$-test).

C   Representative flow cytometry dot plots of CD11b[+]/F4/80[+]/Ly6C[High/Low] and CD11b[+]/Cripto[+]/F4/80[+]/Ly6C[High/Low] cells at days 2, 3, and 5 after injury.

D   Quantification of F4/80[+]/Ly6C[High/Low] cells in the CD11b[+] and CD11b[+]/Cripto[+] cell population from injured muscles at the indicated time points. Data represent mean ± SEM ($n$ = 5 biological replicates; ***$P$ < 0.001, Student's $t$-test).

days 2 and 5 after injury and found that it was comparable in the two groups (Fig 3A), thus providing further evidence that Cripto[My-LOF] did not affect MP accumulation in the injured muscle. To further investigate the nature of these MPs, Cripto[My-LOF] and Control MPs were FACS-isolated by GFP expression, and the expression profile of genes associated with either pro- or anti-inflammatory phenotype analyzed by qRT–PCR at days 2 and 5 after injury. Cripto KO MPs showed a transient increase in the pro-inflammatory marker *Nos2* at day 2, while other pro-inflammatory genes (*Mcp1*, *Tnfα*) were not affected at both time points (Fig 3B). Of note, at day 2, *Arg1* and *Fizz1*, which identify the CD206[+] MPs [24], were similarly expressed in the two groups (Fig 3B), consistent with the equal distribution of the CD206[+] MPs at this time point (Fig 2D). Conversely, later on at day 5 both genes were significantly downregulated in Cripto KO cells, whereas other anti-inflammatory markers (*Il10*, *Il4rα*, and *Tgfβ*) were not affected (Fig 3B). These findings were consistent with

the reduction of the CD206[+] MPs in Cripto[My-LOF] (Fig 2E) and provided molecular support for the idea that modulation of MP phenotypic plasticity is perturbed in the absence of Cripto.

### Cripto modulates TGFβ signaling in the infiltrating MPs

Fine-tuning of the TGFβ signaling plays a key role in shaping MP identity [13,25] and Cripto is a well-known modulator of TGFβ signaling through interaction with TGF-β family ligands [14]. For instance, it is known that Cripto attenuates TGFβ-1 signaling by interfering with its binding to the Act-R II receptors [26]. This raised the possibility that Cripto-dependent modulation of TGFβ signaling might play a role in shaping MP plasticity. We therefore investigated the status of TGFβ signaling in Cripto KO infiltrating MPs by looking at the phosphorylation of the TGFβ effector SMAD3. Immunofluorescence analysis of pSMAD3 and CD206 within the GFP[+] cell population revealed a significant increase in pSMAD3[+] cells in the

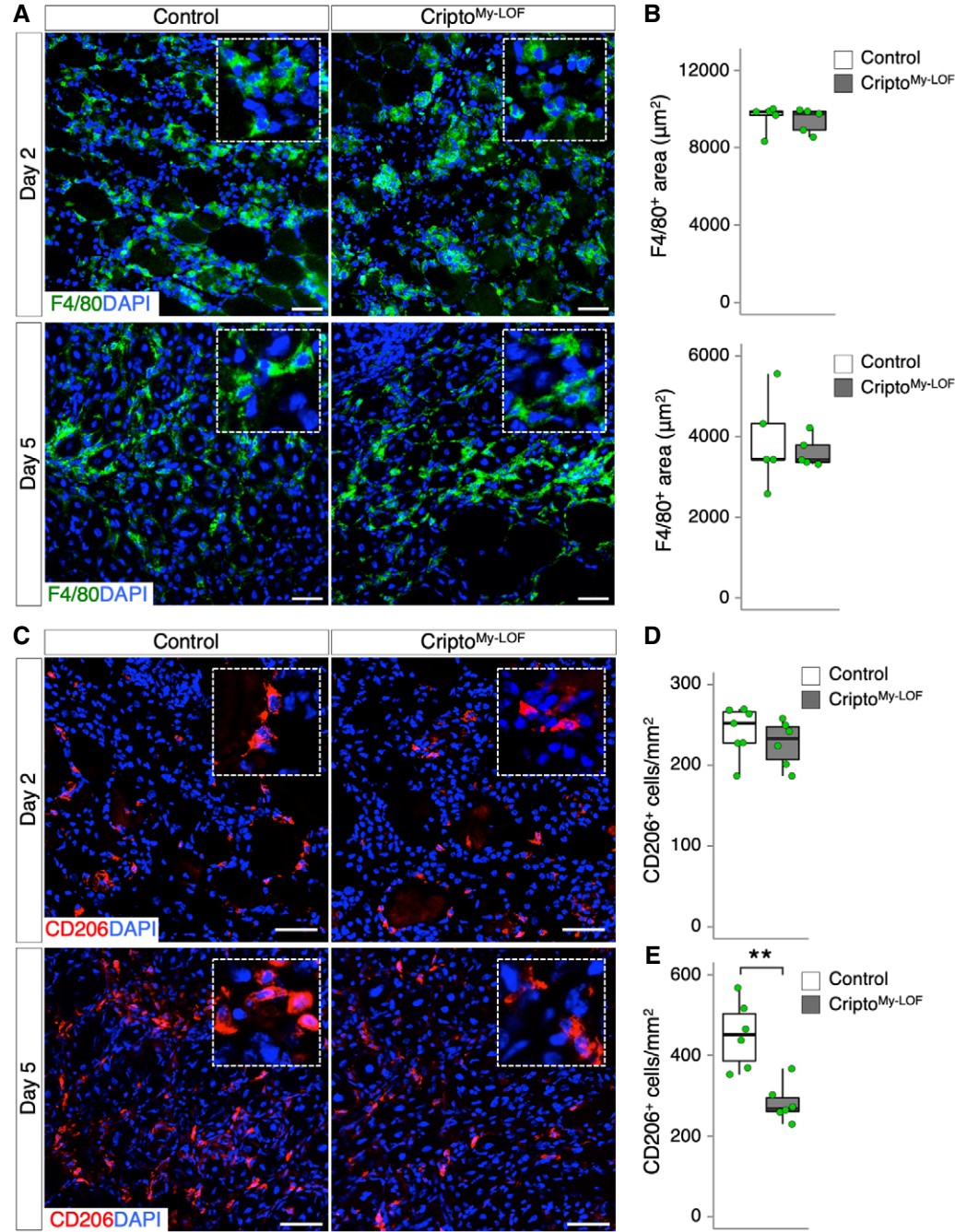

**Figure 2. Myeloid-specific Cripto deletion perturbs MP plasticity without affecting monocyte/macrophage accumulation.**

A    Representative pictures of immunostaining for F4/80 (green) in Control and Cripto^My-LOF TA sections at days 2 (top panel) and 5 (bottom panel) after injury.

B    Quantification of F4/80 staining/damaged area (μm²) at days 2 (top graph) and 5 (bottom graph) after injury.

C    Representative pictures of immunostaining for CD206 in Control and Cripto^My-LOF TA sections at days 2 (top panel) and 5 (bottom panel) after injury, respectively.

D, E    Quantification of CD206⁺ MPs per area (mm²) at days 2 (D) and 5 (E) after injury.

Data information: Nuclei were counterstained with DAPI (blue). Scale bar: 50 μm. Magnification of the boxes is 3.5×. Data are expressed as box plots displaying minimum, first quartile, median, third quartile, and maximum ($n \geq 5$ biological replicates; **$P < 0.01$, Student's $t$-test).

GFP⁺/CD206⁻ population of Cripto^My-LOF mice, which inversely correlated with the decrease in GFP/CD206 double-positive cells (Fig 3C and D). Of note, the expression of both *Tgfβ* and the latent TGFβ binding protein 4 (*LTBP4*), which binds the latent TGFβ promoting its secretion [13], did not increase in Cripto KO MPs (Fig 3B and Appendix Fig S2), indicating that neither expression nor secretion of TGFβ was affected and suggesting that loss of Cripto induced aberrant activation of the signaling. To further explore this

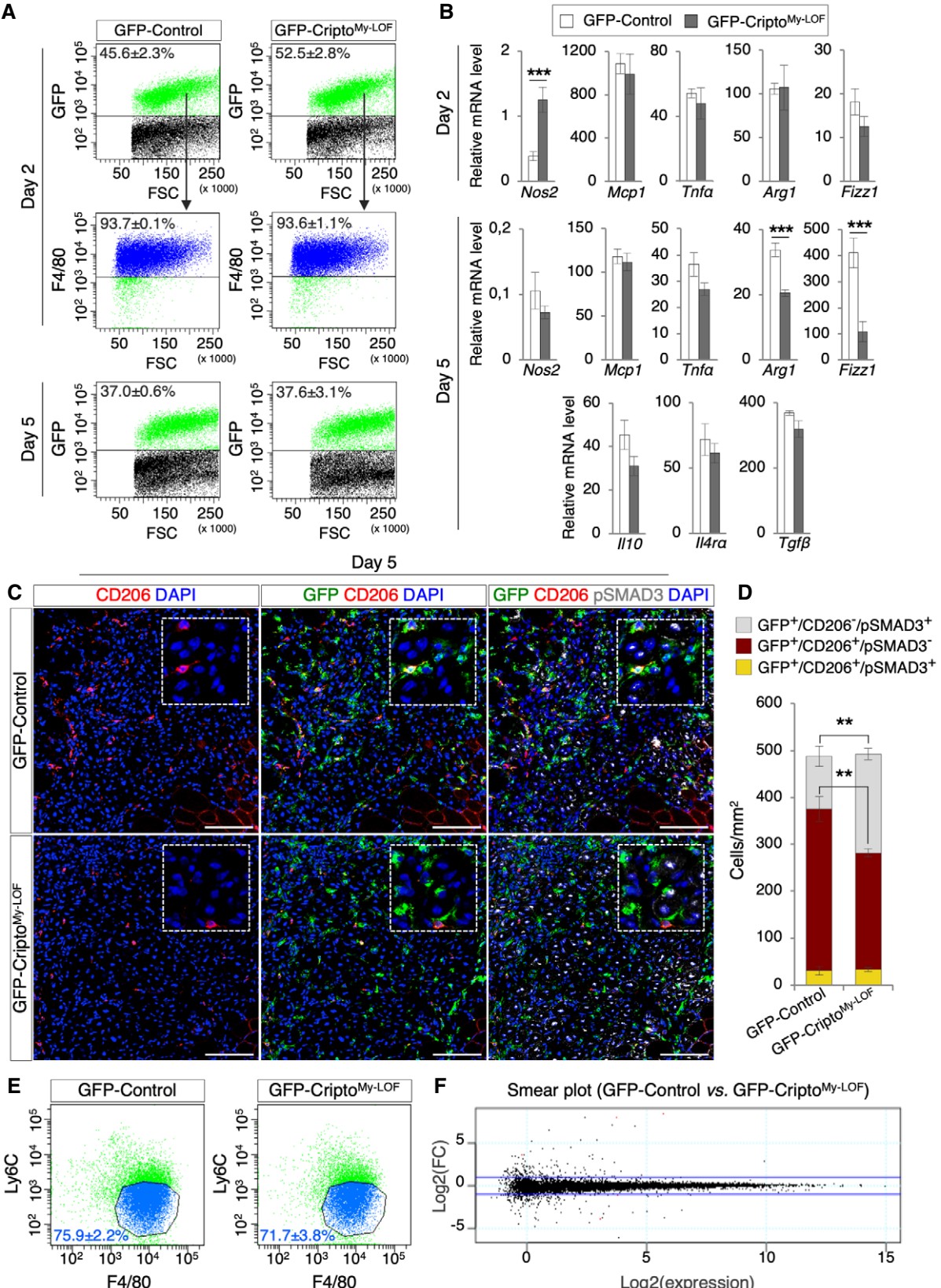

**Figure 3.**

**Figure 3. Cripto controls the proper expansion of CD206⁺ MPs by modulating TGFβ signaling.**

A   Representative flow cytometry dot plots showing the percentage of GFP⁺ cell population in GFP-Control and GFP-Cripto^My-LOF muscles at days 2 (top panel) and 5 (bottom panel) after injury. Percentage of GFP⁺ cells expressing F4/80 in GFP-Control and GFP-Cripto^My-LOF muscles is shown at day 2 after injury (middle panel). Data are mean ± SEM (n = 4 biological replicates; P = ns; Student's t-test).

B   qRT–PCR analysis of pro-inflammatory markers (Nos2, Mcp1, and Tnfα) and anti-inflammatory markers (Arg1, Fizz1, Il10, Il4rα, and Tgfβ) in GFP-Cripto^My-LOF and GFP-Control MPs at days 2 and 5 after injury. Data represent mean ± SEM of relative mRNA level normalized with Gapdh (n ≥ 4 biological replicates; ***P < 0.001, Student's t-test).

C   Representative pictures of immunostaining for GFP (green), CD206 (red), and pSMAD3 (white) in GFP-Control and GFP-Cripto^My-LOF TA sections at day 5 after injury. Nuclei were counterstained with DAPI (blue). Scale bar: 100 μm. Magnification of the boxes is 3.5×.

D   Quantification of GFP⁺/CD206^±/pSMAD3^± cell distribution in TA sections from GFP-Control and GFP-Cripto^My-LOF at day 5 after injury. Nuclei were counterstained with DAPI (blue). Scale bar: 100 μm. Data are mean ± SEM (n = 5 biological replicates; **P < 0.01, Student's t-test).

E   Representative flow cytometry dot plots of F4/80⁺/Ly6C^Low cell population gated on GFP⁺ cells from GFP-Control and GFP-Cripto^My-LOF muscles at day 5 after injury. Data are mean ± SEM (n = 3 biological replicates; P = ns; Student's t-test).

F   Smear plot of RNA-seq data from GFP⁺/F4/80⁺/Ly6C^Low cells FACS sorted from GFP-Control and GFP-Cripto^My-LOF muscles at day 5 after injury. Data show the expression level as log₂ fold change (FC) [log₂(GFP-Control/GFP-Cripto^My-LOF)], against log₂(expression) [log₂(average of gene expression across all samples], for each individual gene. The blue lines correspond to LogFC of 1 and −1 (n = 3 biological replicates).

phenotype and based on the observation that Cripto was preferentially expressed in the anti-inflammatory Ly6C^Low MPs, concomitantly with their accumulation in the regenerating tissue (Fig 1C), we compared the molecular signature of wild-type and Cripto KO Ly6C^Low MPs. Of note, FACS analysis of GFP⁺ cells showed that the percentage of F4/80⁺/Ly6C^Low MPs was comparable in Cripto^My-LOF and Control at day 5 after injury (75.9 ± 2.2% in GFP-Control vs. 71.7 ± 3.8 in GFP-Cripto^My-LOF; Fig 3E), indicating that Cripto deletion did not affect the proper downregulation of Ly6C expression. To further investigate this issue, GFP⁺/Ly6C^Low cells were FACS-isolated from both groups and total RNA was extracted and analyzed by RNA-seq. Smear plot analysis of differentially expressed genes showed no main differences between Control and Cripto KO dataset (Fig 3F and Dataset EV1), indicating that Cripto KO MPs carried the overall expression signature of the Ly6C^Low MPs and suggesting that Cripto was dispensable to induce/maintain the Ly6C^Low phenotype.

Altogether, these findings indicated that Cripto acts as an extrinsic modulator of MP plasticity and suggested that it is required for the proper expansion/maintenance of the CD206⁺ anti-inflammatory MP population.

## Cripto-dependent modulation of MP phenotypes promotes skeletal muscle regeneration in acute injury and disease

We have recently shown that Cripto expression in the infiltrated MPs cannot compensate for the lack of Cripto in the myogenic compartment [19], suggesting that Cripto exerts a cell-type-specific role in the regeneration process. To investigate the effect of Cripto-dependent modulation of MP phenotypes in the repair process, we performed morphometric analysis of Cripto^My-LOF and Control TA muscles at days 5 and 30 after CTX-induced injury (CTX-I) (Fig 4A and B). Quantification of the minimal Feret's diameter of centrally nucleated fibers (CNF) showed no significant differences in the two groups at both time points (Fig 4C), indicating that regeneration was proceeding normally in Cripto^My-LOF mice following a single muscle injury. To investigate this phenotype further and given the incomplete deletion of Cripto in the overall MP population (Fig EV2C), we used a more challenging model of re-injury [27]. To this end, Cripto^My-LOF and Control TA muscles were injected with CTX and allowed to recover for 30 days; regenerated muscles were then re-injured by a second

CTX injection (CTX-II), and the minimal Feret's diameter of CNF was determined at days 5 and 30 after re-injury (Fig 4D and E). Unlike the absence of phenotype observed after a single muscle injury (CTX-I), a significant shift toward smaller regenerating myofibers was detected in Cripto^My-LOF muscles at both days 5 and 30 after re-injury (Fig 4F), which is a clear indication of altered maturation of regenerating myofibers. Furthermore, the CD206⁺ anti-inflammatory MPs significantly decreased in Cripto^My-LOF TA muscles at day 5 after re-injury (Fig EV3A), consistent with that observed in the single injury model (Figs 2E and 3D). Finally, to test whether myeloid Cripto deficiency may affect the satellite cell compartment, we analyzed the distribution of Pax7⁺ satellite cells. A transient reduction in Pax7⁺ cells was observed in Cripto^My-LOF at day 5 after single injury, which was recovered at late time points (Fig EV3B and C).

These findings prompted us to investigate the effect of myeloid Cripto deficiency in disease conditions, such as in the muscle dystrophinopathies. To address this issue directly, we extended the analysis to the mdx mouse, which is a well-characterized model of Duchenne muscular dystrophy (DMD). Of note, while in acute injury a precise sequential presence of pro-inflammatory and anti-inflammatory MPs sustains skeletal muscle repair and regeneration, in DMD, the two MP populations coexist but fail to promote tissue repair and homeostasis recovery, eventually leading to fibrosis and fat deposition [2]. In order to investigate the effect of myeloid-specific Cripto deletion in adult mdx mice, we performed transplantation experiments of bone marrow (mutant or not for myeloid Cripto) into dystrophic mice. Bone marrow from GFP-Cripto^My-LOF and GFP-Control mice was first transplanted into lethally irradiated 4-month-old mdx recipient mice (10⁷ cells/mouse). Sixteen weeks after transplantation, 8-month-old Cripto^My-LOF (mdx-Cripto^My-LOF) and Control (mdx-Control) mice were sacrificed (Fig 5A). FACS analysis showed that a large majority (∼ 80%) of the infiltrated MPs were GFP⁺ in both mdx-Cripto^My-LOF and mdx-Control muscles (Fig EV4A), indicating that, as in acute injury, Cripto deletion did not affect MP accumulation in dystrophic muscles. Interestingly, the large majority of both Control and Cripto^My-LOF GFP⁺ MPs were Ly6C^Low (Fig EV4B). We thus assessed the distribution of CD206⁺ MPs in mdx-Cripto^My-LOF and mdx-Control diaphragms by double staining for GFP and CD206. GFP-positive cells largely infiltrated the diaphragm of both mdx-Cripto^My-LOF and mdx-Control mice

(Fig 5B left panels and C), in line with the idea that this is the most severely affected muscle in *mdx* mice [11]. Remarkably, quantification of the GFP/CD206 double-positive cells revealed a

significant decrease in the CD206$^+$ MPs in *mdx*-Cripto$^{My\text{-}LOF}$ compared to Control (Fig 5B left panels and D). Complementary to these findings, double staining of muscle sections with GFP

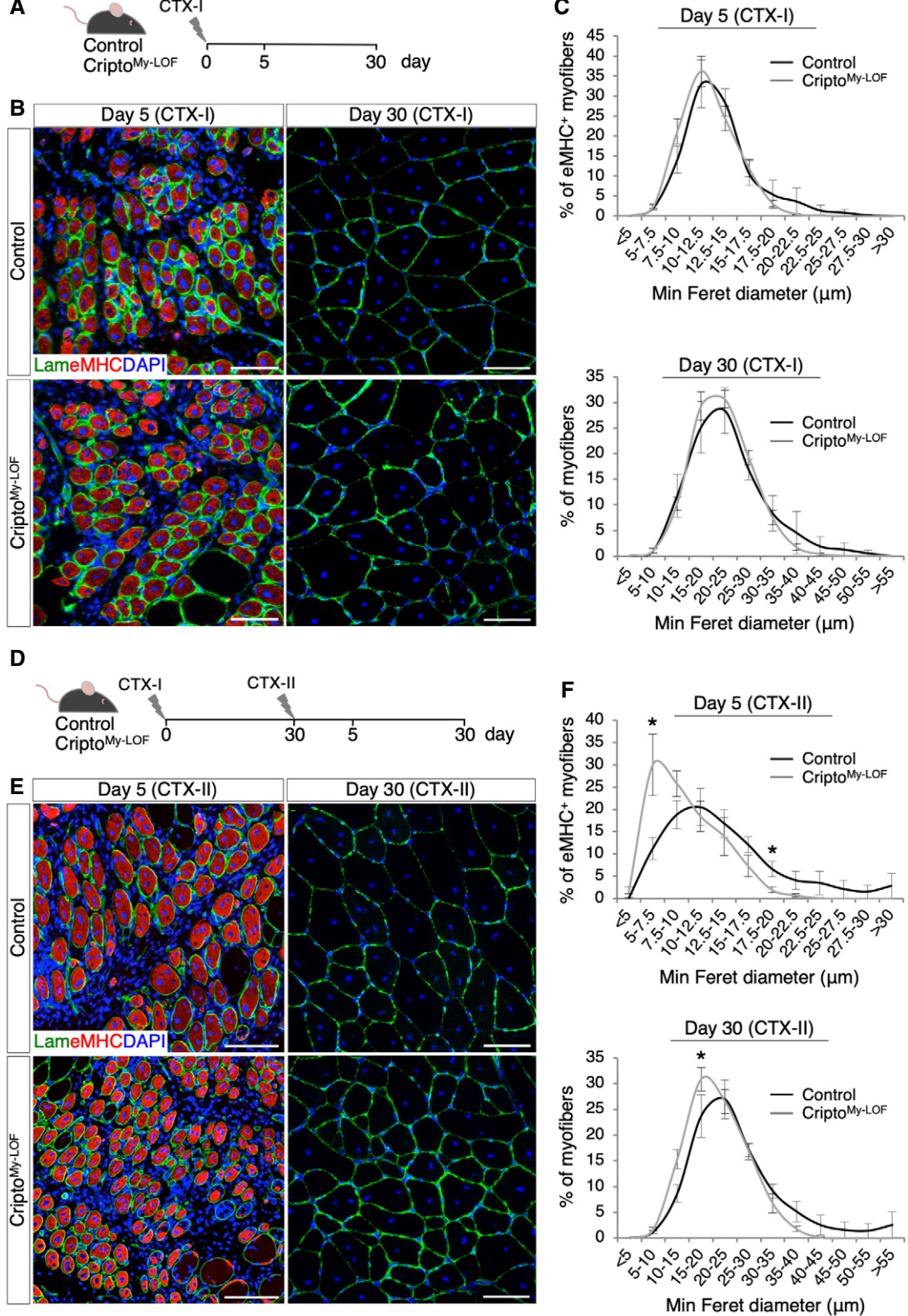

**Figure 4.**

**Figure 4. Effect of myeloid-specific Cripto deletion on skeletal muscle regeneration process.**

A   Schematic representation of the experimental design in the single injury model (CTX-I).
B   Representative pictures of double immunostaining with laminin (green) and embryonic myosin heavy chain (eMHC; red) at days 5 (left panel) and 30 (right panel) after cardiotoxin (CTX) injection.
C   Minimal Feret's diameter distribution of Control and Cripto[My-LOF] centrally nucleated myofibers at days 5 (top panel) and 30 (bottom panel) after CTX injection. Data are mean ± SEM ($n$ = 5 biological replicates; $P$ = ns, Student's $t$-test).
D   Schematic representation of the experimental design in the re-injury model (CTX-II).
E   Representative pictures of double immunostaining with laminin (green) and eMHC (red) at days 5 (left panel) and 30 (right panel) after re-injury (CTX-II).
F   Minimal Feret's diameter distribution of Control and Cripto[My-LOF] centrally nucleated myofibers at days 5 (top panel) and 30 (bottom panel) after re-injury (CTX-II).

Data information: Nuclei were counterstained with DAPI (blue). Scale bar: 100 μm. Data are mean ± SEM ($n$ = 5 biological replicates; *$P$ < 0.05, Student's $t$-test).

and pSMAD3 revealed a positive trend toward increased number of pSMAD3[+] cells in the GFP population (Fig 5B middle and right panels and E), suggesting a tendency to increased TGFβ signaling also in *mdx* Cripto KO MPs. Activation of TGFβ signaling promotes mesenchymal transition and fibrosis and contributes to disease worsening in *mdx* mice [9,13,28]. Consistent with this evidence, quantification of collagen deposition/fibrosis by Picrosirius red staining and hydroxyproline assay showed a significant increase in collagen accumulation in *mdx*-Cripto[My-LOF] diaphragm muscles (Fig 5F–H). Complementary to these findings, minimal Feret's diameter revealed a significant accumulation in *mdx*-Cripto[My-LOF] diaphragm muscle of smaller CNF at the expense of the large fibers (Fig 5I and J). Consistently, increased variance coefficient of minimal Feret's diameter indicated a higher variability of myofiber size in *mdx*-Cripto[My-LOF] mice (Fig 5K). Furthermore, the density of eMHC-positive myofibers was significantly reduced in *mdx*-Cripto[My-LOF] diaphragms (Fig 5L and M), providing further evidence that myeloid-specific *Cripto* deletion reduces the regenerative potential of the *mdx* diaphragm muscle leading to exacerbation of the dystrophic phenotype. Consistent with that observed in acute injury, no significant difference was found in the number of Pax7[+] cells between the two groups (Fig EV4C), thus suggesting that ablation of myeloid Cripto did not affect, at least markedly, the satellite cell compartment.

Collectively, these results indicated that myeloid Cripto sustains accumulation of CD206[+] anti-inflammatory MPs in acutely injured and dystrophic muscles, enhancing muscle regeneration/repair.

## Cripto[My-LOF] affects endothelial plasticity through modulation of EndMT in injured and dystrophic muscle

Despite extensive research efforts, knowledge of the specific contribution of the MP subpopulations to rebuilding the damaged muscle is still limited. It is known that anti-inflammatory MPs produce angiogenic factors and promote neovascularization [29,30]. We therefore investigated the effect of Cripto[My-LOF] on vascular remodeling of injured and dystrophic muscles. Time-course immunofluorescence analysis for CD31 at days 5 and 30 after single CTX injection (CTX-I) showed a significant reduction in both capillary density (CD) and area (CA) in Cripto[My-LOF] muscles (Fig 6A–C), whereas no difference was observed in resting conditions (Appendix Fig S3). Of note, while the reduction in the CD was transient in Cripto[My-LOF] muscles and correlates with defective accumulation of CD206[+] MPs at day 5, reduction in the CA persisted at day 30, when inflammation is resolved (Fig 6B and C). To further support the CA analysis at day 5, a separate quantification of the small and large CA revealed an inverse distribution of the different capillary sizes in the two groups (CA < 20 μm$^2$, 31.5 ± 3.7% in Control vs. 41.4 ± 0.6% in Cripto[My-LOF]; CA > 100 μm$^2$, 11.2 ± 1.3% in Control vs. 4.9 ± 0.6% in Cripto[My-LOF]; Fig 6D). We then extended the analysis to both re-injured (CTX-II) and dystrophic (*mdx*) muscles by staining for CD31 or VE-cadherin (VEcad), respectively (Fig 6E and F). Consistent with the findings above, Cripto[My-LOF] re-injured and dystrophic muscles showed a significant reduction in CA as

**Figure 5. Myeloid-specific Cripto deletion reduced the accumulation of CD206[+] MPs and worsens the *mdx* phenotype.**

A     Schematic representation of the experimental strategy. Four-month-old lethally irradiated *mdx* mice were transplanted with either GFP-Cripto[My-LOF] (*mdx*-Cripto[My-LOF]) or GFP-Control (*mdx*-Control) bone marrow and sacrificed at 16 weeks (wks) after transplantation.
B     Representative pictures of immunostaining of *mdx*-Control and *mdx*-Cripto[My-LOF] diaphragm sections with GFP (green) and CD206 (white; left panel), pSMAD3 (red; middle panel), and GFP and pSMAD3 (right panel).
C–E   Quantification of GFP (C)-, GFP/CD206 (D)-, and GFP/pSMAD3 (E)-positive cells per area (mm$^2$) in *mdx*-Control and *mdx*-Cripto[My-LOF] diaphragms. Data are expressed as box plots displaying minimum, first quartile, median, third quartile, and maximum ($n$ ≥ 5 biological replicates; **$P$ < 0.01, Student's $t$-test).
F, G   Representative pictures of Picrosirius red staining of *mdx*-Control and *mdx*-Cripto[My-LOF] diaphragm sections (F) and quantification of Picrosirius red staining (G). Data are expressed as percentage of stained area and are box plots displaying minimum, first quartile, median, third quartile, and maximum ($n$ = 5 biological replicates; *$P$ < 0.05, Student's $t$-test).
H     Hydroxyproline (HOP) concentration in *mdx*-Control and *mdx*-Cripto[My-LOF] diaphragm muscle sections. Data are expressed as HOP levels (μg) per muscle section volume (mm$^3$) and as box plots displaying minimum, first quartile, median, third quartile, and maximum ($n$ = 5 biological replicates; **$P$ < 0.01, Student's $t$-test).
I–K   Representative images of laminin (red) immunostaining of *mdx*-Control and *mdx*-Cripto[My-LOF] diaphragm sections (I) and minimal Feret's diameter distribution (J) and variance coefficient (VC; K) of centrally nucleated myofibers. Data are expressed as mean ± SEM ($n$ = 5 biological replicates; *$P$ < 0.05; ***$P$ < 0.001; Student's $t$-test) and box plots displaying minimum, first quartile, median, third quartile, and maximum ($n$ = 5 biological replicates; **$P$ < 0.01; Student's $t$-test).
L, M   Representative images of eMHC (red) immunostaining of *mdx*-Control and *mdx*-Cripto[My-LOF] diaphragm sections (L) and quantification of eMHC[+] myofibers per area (mm$^2$, M). Data are expressed as box plots displaying minimum, first quartile, median, third quartile, and maximum ($n$ = 5 biological replicates; *$P$ < 0.05; Student's $t$-test).

Data information: Nuclei were counterstained with DAPI (blue). Scale bar: 100 μm. Magnification of the boxes is 3.5×.

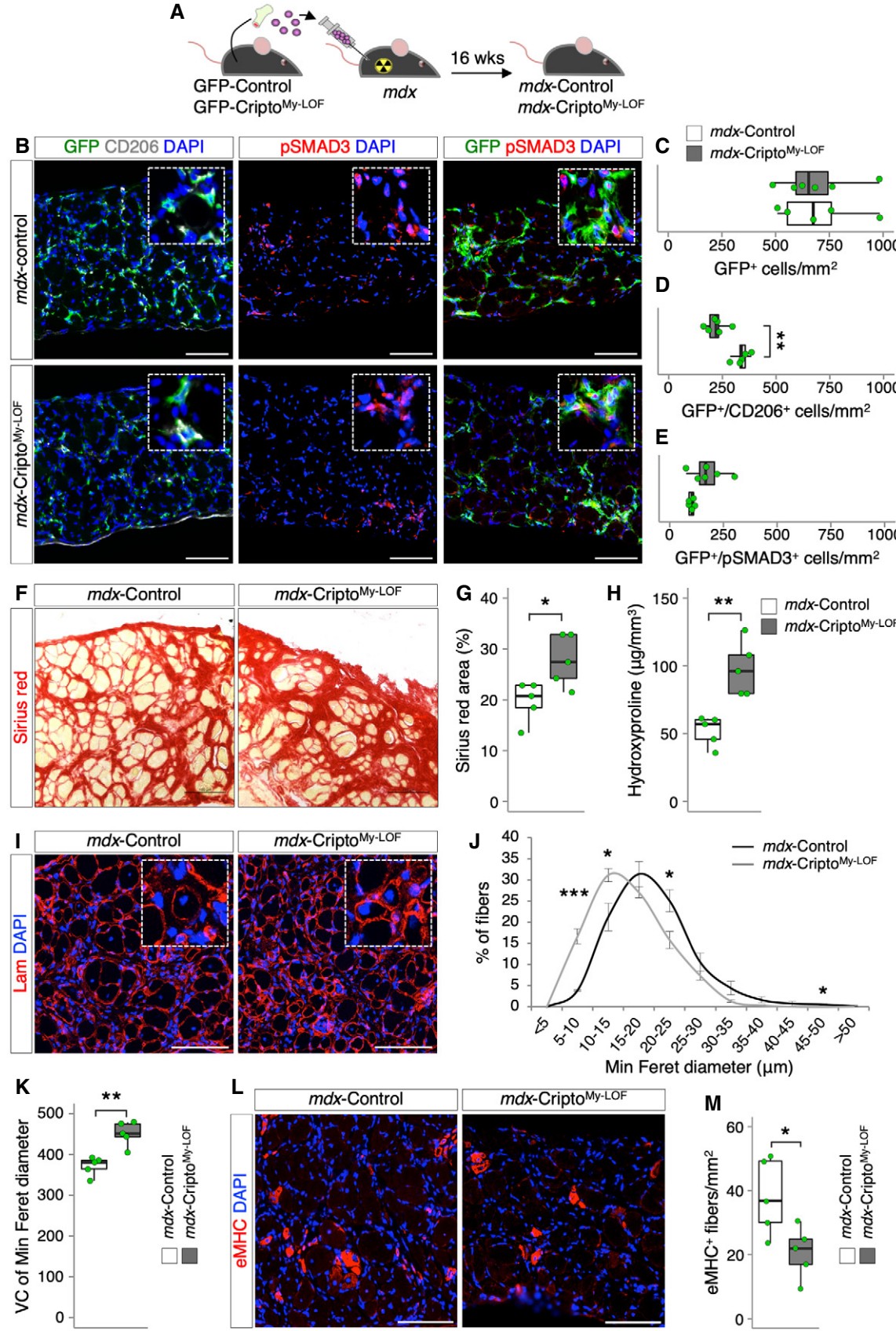

**Figure 5.**

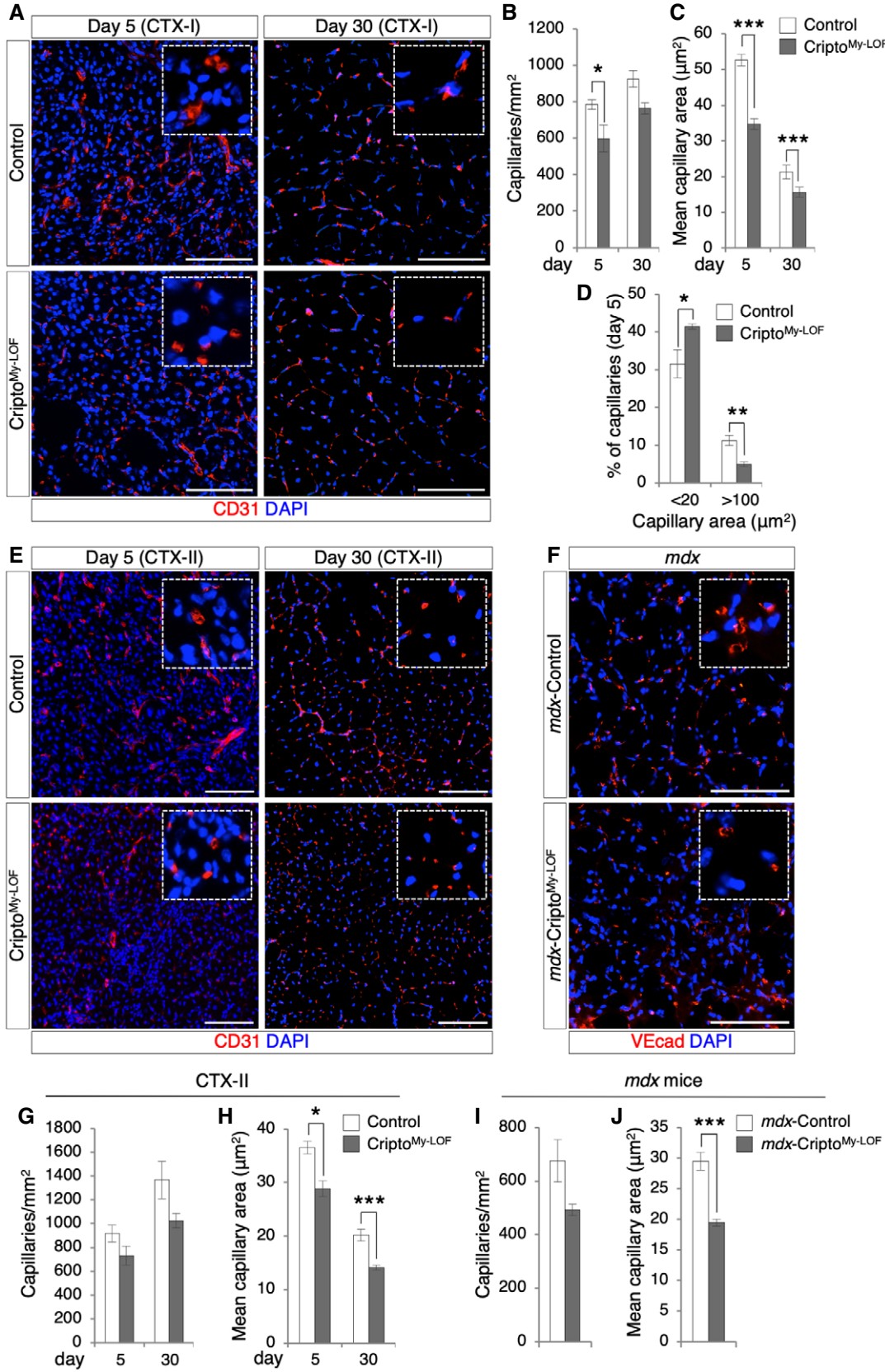

Figure 6.

Figure 6. Myeloid-specific Cripto deletion affects vascular remodeling.

A    Representative pictures of CD31 (red) immunostaining of Control and Cripto[My-LOF] TA sections at days 5 (left panel) and 30 (right panel) after single injury (CTX-I).
B–D  Quantification of CD31[+] capillary number per area (mm²; B), average of capillary cross-sectional area (µm²; C), and percentage of small (< 20 µm²) and large (> 100 µm²) capillaries (D) in Control and Cripto[My-LOF] injured muscles at indicated time points after single injury (CTX-I). Data are mean ± SEM (n = 5 biological replicates; *P < 0.05; **P < 0.01; ***P < 0.001, Student's t-test).
E    Representative pictures of CD31 (red) immunostaining of Control and Cripto[My-LOF] TA sections at days 5 (left panel) and 30 (right panel) after re-injury (CTX-II).
F    Representative pictures of mdx-Control and mdx-Cripto[My-LOF] diaphragm sections immunostained with VEcad (red).
G, H Quantification of CD31[+] capillary number per area (mm²; G) and average of capillary cross-sectional area (µm²; H) in Control and Cripto[My-LOF] injured muscles at the indicated time points after re-injury (CTX-II). Data are mean ± SEM (n = 6 biological replicates; *P < 0.05; ***P < 0.001, Student's t-test).
I, J Quantification of capillary number per area (mm²; I) and average of capillary cross-sectional area (µm²; J) in mdx-Control and mdx-Cripto[My-LOF] diaphragm muscles. Data are mean ± SEM (n = 5 biological replicates; ***P < 0.001, Student's t-test).

Data information: Nuclei were counterstained with DAPI (blue). Scale bar: 100 µm. Magnification of the boxes is 3.5×.

well as a trend toward a decreased CD (Fig 6G–J), providing evidence that Cripto-dependent modulation of CD206[+] anti-inflammatory MPs influenced vascular remodeling.

Emerging evidence points to a key role of infiltrating MPs in regulating the fate of endothelial-derived progenitor cells. Specifically, MPs are crucial to restrict the process of Endothelial-to-Mesenchymal Transition (EndMT) by which endothelial cells (ECs) loose endothelial markers and functions to acquire a mesenchymal phenotype [8,9] leading to vascular abnormalities. To date, the contribution of the different MP populations in the control of EndMT remains largely unclear. Given the predominant effect of Cripto KO on the CD206[+] MPs, we assumed that Cripto[My-LOF] mice are a suitable model to address this issue. We therefore investigated the extent of EndMT in Cripto[My-LOF] and Control mice. To this end, TA muscles at day 5 were double stained with the EC markers (VEcad or CD31) and either the EndMT inducer KLF4 [31] or the mesenchymal markers TCF4 and PDGFRα (Figs 7A and E, and EV5A). Quantification analysis revealed a strong, consistent increase in double-positive VEcad/KLF4 and VEcad/TCF4 (Fig 7B and C), CD31/TCF4 (Fig 7D), and CD31/PDGFRα (Fig 7F) cells in Cripto[My-LOF] mice compared to Control, indicating increased propensity of ECs to undergo EndMT. The increased tendency to acquire mesenchymal gene expression was rather specific of the EC population since the overall number of CD31[−]/PDGFRα[+] cells was comparable between Cripto[My-LOF] and Control (Fig 7G).

Several signaling molecules can induce EndMT, which can be produced either by the resident cells of the injured tissue or by the infiltrated immune cells. One of the best characterized inducers of EndMT is TGFβ [8,32]. We therefore looked at pSMAD3 in the VEcad[+]/KLF4[+] cell population by triple immunostaining of muscle sections for VEcad, KLF4, and pSMAD3 (Figs 7H and EV5B). The distribution of VEcad/KLF4/pSMAD3 triple-positive cells significantly increased in Cripto[My-LOF] muscles (Fig 7I), indicating that increased EndMT positively correlates with increased/persistent activation of TGFβ signaling in ECs. To further investigate the effect of Cripto[My-LOF] on the modulation of vascular remodeling, we extended the analysis to re-injured and dystrophic muscles (Fig 7J and K). Consistent with the above findings, the frequency of VEcad/KLF4 double-positive cells similarly increased in Cripto[My-LOF] re-injured (CTX-II) and dystrophic (mdx-Cripto[My-LOF]) muscles (Fig 7L and M). Of note, we could not observe any significant increase in EndMT in Cripto[My-LOF] muscles at days 2 and 3 after single injury (Fig EV5C and D). Similarly, no difference was found in the CD206[+] MP population between Cripto[My-LOF] and Control at these time points (Fig 2C and D, and Appendix Fig S4). Thus, our

evidence of an invariable inverse correlation between the abundance of CD206[+] MPs and the extent of EndMT pointed to a key role of this MP population in restricting EndMT.

Altogether, these findings provided evidence that Cripto-dependent modulation of the MP phenotype controls vascular remodeling in injured and dystrophic muscles, at least in part, by counteracting TGFβ-induced EndMT.

## Discussion

Macrophage (MP) polarization is a highly dynamic process and it is now clear that the M1/M2 phenotypes only define the extremes of a phenotypic continuum, with MPs often displaying characteristics of both states at the same time in vivo [33]. The molecular basis of this plasticity is still far to be elucidated also due to the fact that in vitro polarization poorly mimics the complexity of in vivo conditions, including for instance the influence of microenvironment. In the present study, we reveal a key role of the extracellular membrane protein Cripto in shaping MP phenotypes in injured and dystrophic muscles. Specifically, we used the intramuscular injection of cardiotoxin (CTX) as a model of acute muscle injury. Although the CTX injection is not physiological and the muscle damage is different to that induced by exercise or chronic disease, it is a widely used model since it induces a sequential and synchronous skeletal muscle regeneration characterized by neutrophil and macrophage infiltration [27]. We provide unprecedented evidence that Cripto is predominantly expressed at the membrane of anti-inflammatory (Ly6C[Low]) MPs, which progressively infiltrates the muscle following CTX-induced acute injury. In line with its expression profile, myeloid-specific Cripto genetic ablation (Cripto[My-LOF]) reduces the number of CD206[+] anti-inflammatory MPs both in acute injury and in disease.

Molecular and immunofluorescence analysis indicate that while the CD206[+] MPs are present in normal number in Cripto[My-LOF] muscles at early time point after acute injury, they are significantly reduced later on, concomitantly to the time of their increase in the regenerating muscle. It thus stands to reason that this anti-inflammatory MP population is induced but then fail to properly expand/accumulate in the absence of Cripto. Since myeloid-specific Cripto KO does not affect macrophage accumulation, it is reasonable to assume that Cripto[My-LOF] MPs show a mixed phenotype. In line with this idea, while the expression of key anti-inflammatory genes (Arg1 and Fizz1) is reduced in Cripto KO MPs at day 5 after injury, consistent with the reduced CD206[+] MPs, other anti-inflammatory-specific

genes, including *Il10*, *Il4ra*, and *Tgfß*, are not affected, pointing to perturbation of MP plasticity rather than impaired phenotypic transition.

Recent findings suggest that at early stage of acute injury, infiltrating MP populations are broadly activated rather than polarized, as they express both pro- and anti-inflammatory genes [34]. Our

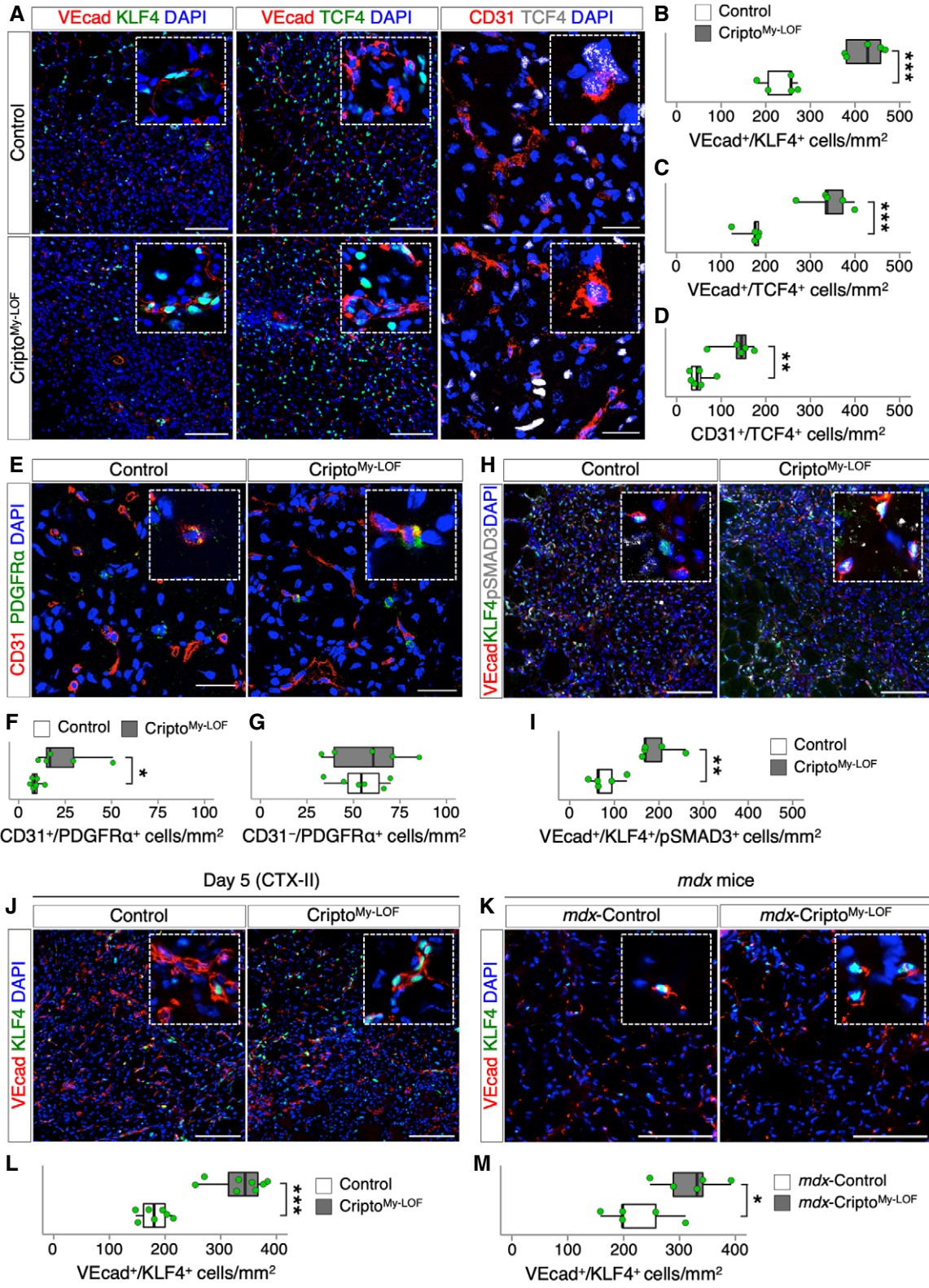

**Figure 7.**

**Figure 7. Cripto modulates endothelial plasticity by restricting Endothelial-to-Mesenchymal Transition.**

A     Representative pictures of double immunostaining with VEcad (red; left and middle panels) and KLF4 (green, left panel) or TCF4 (green, middle panel), and with CD31 (red; right panels, confocal pictures) and TCF4 (white, right panel) on Control and Cripto[My-LOF] TA sections at day 5 after injury.

B–D   Quantification of VEcad/KLF4 (B), VEcad/TCF4 (C), and CD31/TCF4 (D) double-positive cells per area (mm$^2$) in Control and Cripto[My-LOF] TA sections at day 5 after injury.

E     Representative Confocal pictures of CD31$^+$ (red) and PDGFRα$^+$ (green) immunostaining on Control and Cripto[My-LOF] TA sections at day 5 after injury.

F, G  Quantification of CD31$^+$/PDGFRα$^+$ (F) and CD31$^-$/PDGFRα$^+$ (G) cells per area (mm$^2$).

H, I  Representative pictures of triple immunostaining with VEcad (red), KLF4 (green), and pSMAD3 (white) on Control and Cripto[My-LOF] TA sections at day 5 after injury (H) and quantification of VEcad/KLF4/pSMAD3 triple-positive cells per area (mm$^2$; I).

J, K  Representative pictures of double immunostaining with VEcad (red) and KLF4 (green) on Control and Cripto[My-LOF] TA sections at day 5 after re-injury (CTX-II; J) and on diaphragm sections of *mdx*-Control and *mdx*-Cripto[My-LOF] (K).

L, M  Quantification of VEcad/KLF4 double-positive cells per area (mm$^2$) in Control and Cripto[My-LOF] TA sections at day 5 after re-injury (CTX-II; L) and in diaphragm sections of *mdx*-Control and *mdx*-Cripto[My-LOF] (M).

Data information: Nuclei were counterstained with DAPI (blue). Scale bar: 100 μm. Confocal pictures scale bar: 25 μm. Magnification of the boxes is 3.5×. Data are expressed as box plots displaying minimum, first quartile, median, third quartile, and maximum ($n \geq 5$ biological replicates; *$P < 0.05$; **$P < 0.01$; ***$P < 0.001$; Student's *t*-test).

findings are in line with this assumption; for instance, the anti-inflammatory genes *Arg1* and *Fizz1* are already expressed at day 2 after injury. It is thus tempting to speculate that Cripto expression at the early time point may also reflect the broad activation of the infiltrated MPs. Interestingly, at this early stage of acute injury, *Cripto* genetic ablation affects neither pro- nor anti-inflammatory gene expression, with the exception of the pro-inflammatory marker *Nos2*. Conversely, several lines of evidence indicate that Cripto controls the proper distribution of CD206$^+$ MPs in acute injury and disease. First, the number of CD206$^+$ MPs is consistently reduced in Cripto[My-LOF] injured and dystrophic muscles. Furthermore, Cripto[My-LOF] affects neither the frequency of the GFP$^+$/F4/80$^+$/Ly6C[Low] anti-inflammatory MPs nor their genome-wide expression profile, providing support to the idea that Cripto is dispensable for the induction and maintenance of the Ly6C[Low] MP population, but it is rather required for the maintenance and/or expansion of the CD206$^+$ MP phenotype.

MPs change their phenotypes in response to different stimuli in the microenvironment, among which TGFβ plays a crucial role. Indeed, MPs secrete high level of latent TGFβ protein, which becomes activated [13], and mediate the crosstalk with other muscle cellular components including ECs, fibroblasts, and fibro-adipogenic precursors (FAPs) [9]. TGFβ signaling is modulated via the control of a large repertoire of extracellular and membrane proteins, which regulate the strength of the signaling and the cell-specific response, including Cripto [35]. Indeed, Cripto can attenuate TGFβ-1 signaling by interfering with its binding to the Act-R II receptors [26] thus raising the possibility that Cripto-dependent control of MP plasticity may occur, at least in part, through modulation of TGFβ/Smad signaling. In line with this hypothesis, we find a strong increase in pSMAD3$^+$ cells in the CD206$^-$ MP population in Cripto KO mice, which inversely correlates with the reduction in the CD206$^+$ MPs. The inverse correlation between the number of pSMAD3$^+$ and CD206$^+$ MPs suggests that a persistent/aberrant activation of TGFβ/Smad signaling prevents, at least in part, the proper expansion of the CD206$^+$ anti-inflammatory MPs in the absence of Cripto. This Cripto-dependent modulation of MP plasticity contributes to efficient regeneration/maturation of muscle fibers both in acute injury and in chronic disease conditions. We thus propose that, while myeloid Cripto is dispensable for macrophage polarization, it contributes, along with other factors, to create the proper inflammatory microenvironment for muscle repair and regeneration.

In line with the idea that Cripto exerts a cell-type-specific role in muscle repair [19], myeloid Cripto ablation does not significantly affect the satellite cell number, suggesting the involvement of other cell types. It is known that anti-inflammatory MPs exert a key role in the rebuilding of skeletal muscle by producing anti-inflammatory cytokines and by secreting pro-angiogenic factors that promote vascular remodeling. Furthermore, MPs play a crucial role in shaping the fate of endothelial-derived progenitor cells by restricting the process known as Endothelial-to-Mesenchymal Transition (EndMT) by which ECs loose endothelial markers and functions and acquire a mesenchymal phenotype [8–10]. Several lines of evidence suggest that this phenotypic switch likely occurs as a continuum with temporal changes in the expression of endothelial and mesenchymal markers [36]. How this multistep process is controlled, and whether intermediate/plastic phenotypes can be maintained without progressing toward a fully mesenchymal state is a major unresolved issue. So far, the contribution of MPs to the process of EndMT in skeletal muscle, and their functional interaction with other muscle cell population, has been investigated by using chemical/toxin-induced depletion of MPs [4,8,10,37]. Thus, the specific contribution of the different and occasionally antagonistic MP phenotypes to this process remains poorly understood. Here, we provide evidence that the CD206$^+$ MPs exert a key role in regulating EndMT both in injured and dystrophic muscles. Specifically, we show that the reduction in the CD206$^+$ MPs invariably correlates with increased EndMT in Cripto[My-LOF] mice, pointing to a key role of Cripto in this complex scenario. It is reasonable to assume that Cripto may exert its activity either directly, as a paracrine factor produced by the anti-inflammatory MPs, or indirectly through regulating maintenance/expansion of CD206$^+$ MP population. The invariable inverse correlation between the CD206$^+$ MPs and the extent of EndMT would favor the hypothesis that a proper expansion/maintenance of these anti-inflammatory MPs is a crucial condition to prevent excessive EndMT.

Increased TGFβ signaling is major contributor to excessive EndMT and fibrosis, and predictor of chronic muscle disease worsening [12]. Yet, the specific contribution of the pro- and anti-inflammatory MPs to this scenario is still a question of debate and data are somehow controversial. Here, we show that reduced accumulation of CD206$^+$ MPs associates with increased EndMT and fibrosis in *mdx*-Cripto[My-LOF] mice. Although we cannot assume any causative correlation between increased EndMT and fibrosis, we suggest that

accumulation/expansion of CD206$^+$ MPs contributes to myofiber regeneration/maturation in dystrophic mice, preventing excessive collagen deposition. In line with this idea, it has been recently showed that anti-inflammatory MPs are associated to areas with regenerating myofibers in *mdx* mice [13].

Collectively, these findings point to a previously unknown role for Cripto as a regulator of MP plasticity and EndMT in skeletal muscle acute injury and disease. The human relevance of our findings remains to be determined. Indeed, human and mouse immune system exhibit extensive similarities; however, some differences have been reported [38]. For instance, how the continuum of human and murine macrophages phenotypes is aligned is still unclear [39]. Thus, although differences may exist between mouse and human MP phenotypes, Cripto$^{My-LOF}$ mice provide a unique genetic model to study the specific contribution of CD206$^+$ anti-inflammatory MP population in different pathologies where inflammation play a critical role, including vascular disease and cancer.

# Materials and Methods

### Mouse models and genotyping

Experiments were done in accordance with the law on animal experimentation (article 7; D.L. 116/92) under the Animal Protocol approved by the Department of Public Health, Animal Health, Nutrition and Food Safety of the Italian Ministry of Health. Bone marrow transplantation experiments were approved by the ethics committee of the Barcelona Biomedical Research Park (PRBB) and by the Catalan Government.

Experimental mice were 10–12 weeks old, if not differently specified. We did not perform randomization for blinded experiments in this animal study. The *Tg:LysMCre::Cripto$^{fl/fl}$* mice (Cripto$^{My-LOF}$) were generated by crossing the *Tg:LysMCre* strain (Jackson stock number 004781) [22] to *Tg:Cripto$^{fl/fl}$* transgenic mice [21]. *Tg:LysMCre::R26$^{mTmG}$::Cripto$^{fl/fl}$* (GFP-Cripto$^{My-LOF}$) mice were obtained by crossing the *Tg:LysMCre::Cripto$^{fl/fl}$* mice to the *Tg:R26$^{mTmG}$* reporter line [23]. Mice genotyping was performed by PCR using primers listed in Appendix Table S1. To quantify the efficiency of *Cripto* deletion, quantitative real-time PCR (qPCR) of wild-type *Cripto* allele was performed on genomic DNA samples. The amount of wild-type *Cripto* was measured by using specific primers that amplify a DNA region encompassing exon 4, inside the LoxP-flanked region. Sequence-specific primers were designed to amplify exon 2, outside the LoxP-flanked region, that was used as reference PCR. The amount of wild-type *Cripto* was calculated over the reference PCR. Primers are listed in Appendix Table S1.

### Nucleic acid extraction and quantitative RT–PCR analysis

Genomic DNA was isolated by a standard phenol–chloroform method using an SDS/NaCl extraction buffer (200 mM Tris–HCl pH 7.5, 25 mM EDTA, 250 mM NaCl, and 0.5% SDS).

Total RNAs were isolated in TRIzol (Sigma-Aldrich) by using the Direct-Zol Miniprep Kit (Zymo Research) following manufacturer's instructions, and reverse transcribed using QuantiTect Reverse Transcription Kit (Qiagen). Quantitative real-time PCR (qPCR) was performed using SYBR GREEN PCR Master Mix (FluoCycle II™ SYBR®, EuroClone). Primers are listed in Appendix Table S2.

### Muscle injury, histological, and immunofluorescence analysis

To induce muscle damage, 20, 30 μl, or 50 μl of cardiotoxin (CTX; Latoxan, 10 μM) was injected in the tibialis anterior (TA), gastrocnemius, and quadriceps muscle, respectively. Muscles were collected and embedded in OCT (Bio-Optica) or directly snap-frozen in liquid nitrogen-cooled isopentane for cryosection, as previously described [40].

Immunofluorescence (IF) staining was performed on serial muscle sections (10 μm). After fixation in PFA 4% (wt/vol), muscle sections were permeabilized with 0.2% Triton X-100 (Sigma), 1% BSA in PBS for 30 min at room temperature and then blocked in 10% serum, 1% BSA in PBS for 60 min, before incubation with the primary antibody for overnight at 4°C. Appropriate fluorochrome-conjugated antibodies were used as second-step reagent. For Pax7 visualization, muscle sections were processed as previously described [41]. Briefly, cryosections were fixed in 4% (wt/vol) PFA and permeabilized in ice-cold methanol for 6 min at −20°C. Slides were subjected to antigen retrieval in boiling 10 mM Na-citrate (pH 6.0) solution for 15 min, followed by cooling for 30 min. Sections were then incubated in blocking buffer with 4% IgG-free BSA (Jackson) in PBS for 2 h, and endogenous immunoglobulins were blocked with AffiniPure Fab Fragment Goat Anti-Mouse IgG (Jackson) for 30 min at room temperature. Primary antibody was incubated overnight at 4°C. Biotin-conjugated secondary antibody and Alexa Fluor 488-conjugate streptavidin (Invitrogen; S32354) were used for the detection. Nuclei were counterstained with DAPI (1 μg/ml; Roche), and FluorSave Reagent (Calbiochem) was used for mounting. Primary and secondary antibodies used are listed in Appendix Table S3.

### Images acquisition and analyses

Bright-field and fluorescent images were taken by using DM6000B (Leica), ECLIPSE NI-E (Nikon), or Confocal TCS SP5 (Leica) microscopes. Immunofluorescence images showing double and triple fluorescence were acquired separately by using appropriate filters. The different layers were then merged and edited by using Adobe Photoshop CS6. Histological quantifications were performed considering all regenerating areas (20× magnification or 40× magnification).

For quantifying the F4/80-positive area, images were processed through the threshold tool of the ImageJ software to segment out and measure the staining areas. For quantification of CD31- or VEcad-positive staining, the signal from CD31 or VEcad immunostaining was processed by using the ImageJ software as previously described [42]. An empirically determined threshold was used to segment out CD31- or VEcad-positive regions, thereby calculating the density and area of capillaries by using the Analyze Particles tool of the ImageJ software.

Morphometric analyses were performed on sections collected from similar regions of each muscle. Minimal Feret's diameter of muscle fibers was determined on immunostained sections (500–1,000 fibers/muscle) using the ImageJ software. Variance coefficient of minimal Feret's diameter was calculated as reported [43].

For collagen quantification, 10-μm-thick sections were stained with Sirius Red (Sigma-Aldrich) according to standard procedures. Briefly, sections were incubated with 0.1% Sirius Red solution dissolved in aqueous saturated picric acid for 1 h, washed rapidly in 2% acetic acid, dehydrated, and mounted with DPX Mounting. Collagen content was calculated as a percentage of the red-stained area/section using ImageJ (https://imagej.nih.gov/ij/docs/examples/stained-sections/).

### Florescence-activated cell sorting

Muscles were removed and separated from the bones, and dissociated by enzymatic digestion using 3 U/ml Dispase II (Roche), 0.5 U/ml Collagenase A (Roche), 10 μg/ml DNAse I (Roche), 400 μM $CaCl_2$, and 5 μM $MgCl_2$ in phosphate-buffered saline (PBS), for 75 min in a shaking bath at 37°C. Cells were then washed with DMEM (Gibco) and filtered with 100- and 70-μm cell strainers (Corning). Following filtration, red blood cells were removed using BD Pharm Lyse™ (BD Biosciences) according to the manufacturer's instructions, and the remaining cells were re-suspended in PBS 2% FBS (FACS buffer). Macrophages were blocked using purified *Rat Anti-Mouse CD16/CD32* (*Mouse BD Fc Block™*; 1:100) for 15 min at 4°C and then incubated with the indicated primary antibodies for 30 min at 4°C. After washing with ice-cold PBS, cells were either analyzed or sorted with a FACSAria II (BD Biosciences); data were analyzed with FACSDiva software (BD Biosciences). Free channel was used to exclude auto-fluorescent cells from the analysis. Primary antibodies used are listed in Appendix Table S3.

### ELISA

To quantify Cripto protein, 96-well plate was coated with mouse anti-Cripto antibody (100 μl, 0.5 μg/ml) in PBS (pH 7.5) overnight at 4°C and washed three times with PBS-Tween. Unbinding sites were blocked with 1% PBS-BSA (180 μl/well) for 3 h at room temperature (RT). After three washing, protein extracts obtained from $1 \times 10^6$ FACS-sorted $GFP^+$ cells were diluted in PBS, 0.1% BSA, 5 mM EDTA, 0.004% Tween-20 (PBET) with a final volume of 100 μl and incubated overnight at 4°C. Recombinant mouse Cripto (R&D Systems) was used to obtain a standard curve concentration. After washing, the plate was incubated with anti-Cripto biotinylated antibody (1 μg/ml; R&D Systems) in PBET for 1 h at 37°C and then for 1 h at RT. Finally, the plate was incubated for 1 h at RT with avidin/streptavidin complex conjugated with HRP (VECTASTAIN Elite ABC Kit; Vector Laboratories), and developed with o-phenylenediamine peroxidase substrate (Sigma-Aldrich). Absorbance was read at 490 nm on a Benchmark Microplate Reader (Bio-Rad Laboratories).

### Hydroxyproline assay

To determine hydroxyproline (HOP) content in diaphragm muscles, two 10-μm-thick muscle sections/mouse were processed and analyzed using the Hydroxyproline Colorimetric Assay (Sigma-Aldrich; MAK008-1KT), according to the manufacturer's instructions. Absorbance was read at 560 nm on a Synergy HT Microplate Reader (BioTek). HOP levels were normalized by the volume of muscle sections to yield μg/mm³.

### Whole-genome expression analysis

Sequencing read quality was assessed with FastQC [44]. Available online at http://www.bioinformatics.babraham.ac.uk/projects/fastqc).

Over-represented sequences were trimmed with Cutadapt 1.7.1 (http://journal.embnet.org/index.php/embnetjournal/article/view/200), which also discarded reads that were shorter than 30 bp. The resulting reads were mapped against the mouse transcriptome (GRCm38, release 91; dec2017 archive) and quantified using RSEM v1.2.20 [45]. Data were then processed with a differential expression analysis pipeline that used Bioconductor package LIMMA [46] for normalization and differential expression testing. Finally, to visualize the differential expression of each individual gene, $log_2$ average of gene expression level and $log_2$ fold change between Control and KO condition have been reported, respectively, on the *x*-axis and *y*-axis of smear plot.

### Bone marrow transplantation

For bone marrow transplantation, 10-wk-old GFP-Cripto^My-LOF or GFP-Control mice were used as donors. Bone marrow of donor mice was flushed from femur and tibia bones, and $\sim 1 \times 10^7$ cells were transplanted into lethally irradiated C57BL/10ScSn-Dmd^mdx/J (*mdx*) 4-mo-old recipient mice. No-transplanted animals were used as control of irradiation, which generally die 7–21 days after the irradiation.

### Statistical analysis

The exact sample size for each experiment is reported as "*n*" in the caption of each figure. Animals or samples were not randomized during experiments and not excluded from analyses. The investigators were not blinded to group allocation during experiments and outcome analyses. Statistical analyses were performed using unpaired two-sided Student's *t*-tests, and $P < 0.05$ was considered statistically significant. Results are presented as the mean ± SEM (standard error of the mean) or as box plot displaying minimum, first quartile, median, third quartile, and maximum. Box plots were generated using RStudio software (https://www.rstudio.com/).

# Data availability

The dataset produced in this study is available in the following database: Gene Expression Omnibus GSE142072 (https://www.ncbi.nlm.nih.gov/geo/query/acc.cgi?acc=GSE142072).

**Expanded View** for this article is available online.

### Acknowledgements

We thank the Animal House, Integrated Microscopy, and FACS Facilities of IGB-CNR Naples. We kindly thank Prof. Michael Shen for providing the Cripto flx/flx mice. This work is supported by AFM 21534, SATIN-POR Campania FESR 2014/2020, Italian Ministry of Education-University-Research (CTN01_00177 Cluster ALISEI_IRMI; PRIN 2017XJ38A4) and AIRC (IG20736) to GM; and by the Spanish Ministry of Science and Innovation, Spain [grants SAF2015-67369-R, RTI2018-096068-B-I00 and SAF2015-70270-REDT, a María de Maeztu Unit of

Excellence award to UPF (MDM-2014-0370), and a Severo Ochoa Center of Excellence award to the CNIC (SEV-2015-0505)], the UPF-CNIC collaboration agreement, ERC-2016-AdG-741966, La Caixa-HEALTH (HR17-00040), MDA, UPGRADE-H2020-825825, AFM 653, and DPP-E.

## Author contributions

FI, OG, ASc, GA, and FE performed the experiments. FI, OG, ASe, EP, SB, PM-C, and GM designed the work, analyzed, and interpreted the data. FI and GM wrote the manuscript. PM-C, ASe, EP, and OG critically reviewed and revised the manuscript.

## Conflict of interest

The authors declare that they have no conflict of interest.

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
