## [Review Process File · EMBO Reports]

Cripto shapes macrophage plasticity and restricts EndMT in injured and diseased skeletal muscle

Francescopaolo Iavarone, Ombretta Guardiola, Alessandra Scagliola, Gennaro Andolfi, Federica Esposito, Antonio Serrano, Eusebio Perdiguero, Silvia Brunelli, Pura Muñoz-Cánoves and Gabriella Minchiotti

Review timeline:

Submission date:	14 August 2019
Editorial Decision:	1 October 2019
Revision received:	23 December 2019
Editorial Decision:	30 January 2020
Revision received:	6 February 2020
Accepted:	10 February 2020

Editor: Achim Breiling

Transaction Report:

1st Editorial Decision

1 October 2019

Thank you for the transfer of your research manuscript to EMBO reports. We have now received reports from the three referees that were asked to evaluate your study, which can be found at the end of this email.

As you will see, all three referees think that the findings are of interest, but they also have several comments, concerns and suggestions, indicating that a major revision of the manuscript is necessary to allow publication in EMBO reports. As the reports are below, and I think all points need to be addressed, I will not detail them here.

Given the constructive referee comments, we would like to invite you to revise your manuscript with the understanding that all referee concerns must be addressed in the revised manuscript and in a detailed point-by-point response. Acceptance of your manuscript will depend on a positive outcome of a second round of review. It is EMBO reports policy to allow a single round of revision only and acceptance or rejection of the manuscript will therefore depend on the completeness of your responses included in the next, final version of the manuscript.

Revised manuscripts should be submitted within three months of a request for revision; they will otherwise be treated as new submissions. Please contact me if a 3-months time frame is not sufficient, so that we can discuss the revisions further.

When submitting your revised manuscript, please also carefully review the instructions that follow below. Failure to include requested items will delay the evaluation of your revision. When submitting your revised manuscript, we will require:

1) a .docx formatted version of the final manuscript text (including legends for main figures, EV figures and tables), but without the figures included. Please make sure that the changes are highlighted to be clearly visible. Figure legends should be compiled at the end of the manuscript text.

2) individual production quality figure files as .eps, .tif, .jpg (one file per figure), of main figures and EV figures. Please upload these as separate, individual files upon re-submission.

The Expanded View format, which will be displayed in the main HTML of the paper in a collapsible format, has replaced the Supplementary information. You can submit up to 5 images as Expanded View. Please follow the nomenclature Figure EV1, Figure EV2 etc. The figure legend for these should be included in the main manuscript document file in a section called Expanded View Figure Legends after the main Figure Legends section. Additional Supplementary material should be supplied as a single pdf labeled Appendix. The Appendix should have page numbers and needs to include a table of content on the first page (with page numbers) and legends for all content. Please follow the nomenclature Appendix Figure Sx, Appendix Table Sx etc. throughout the text, and also label the figures and tables according to this nomenclature.

For more details please refer to our guide to authors:

See also our guide for figure preparation:

http://wol-prod-cdn.literatumonline.com/pb-assets/embosite/EMBOPress_Figure_Guidelines_061115-1561436025777.pdf

4) a complete author checklist, which you can download from our author guidelines (<https://www.embopress.org/page/journal/14693178/authorguide>). Please insert page numbers in the checklist to indicate where the requested information can be found in the manuscript. The completed author checklist will also be part of the RPF.

Please also follow our guidelines for the use of living organisms, and the respective reporting guidelines: <http://www.embopress.org/page/journal/14693178/authorguide#livingorganisms>

5) that primary datasets produced in this study (e.g. the whole-genome expression analysis) are deposited in an appropriate public database. See:

<http://embor.embopress.org/authorguide#datadeposition>

The accession numbers and database should be listed in a formal "Data Availability" section (placed after Materials & Methods) that follows the model below. Please note that the Data Availability Section is restricted to new primary data that are part of this study.

Data availability

6) We strongly encourage the publication of original source data with the aim of making primary data more accessible and transparent to the reader. The source data will be published in a separate source data file online along with the accepted manuscript and will be linked to the relevant figure.

If you would like to use this opportunity, please submit the source data (for example scans of entire gels or blots, data points of graphs in an excel sheet, additional images, etc.) of your key experiments together with the revised manuscript. If you want to provide source data, please include size markers for scans of entire gels, label the scans with figure and panel number, and send one PDF file per figure.

7) Our journal encourages inclusion of *data citations in the reference list* to directly cite datasets that were re-used and obtained from public databases. Data citations in the article text are distinct from normal bibliographical citations and should directly link to the database records from which the data can be accessed. In the main text, data citations are formatted as follows: "Data ref: Smith et al, 2001" or "Data ref: NCBI Sequence Read Archive PRJNA342805, 2017". In the Reference list, data citations must be labeled with "[DATASET]". A data reference must provide the database name, accession number/identifiers and a resolvable link to the landing page from which the data can be accessed at the end of the reference. Further instructions are available at:

8) Regarding data quantification and statistics, can you please specify, where applicable, the number "n" for how many independent experiments (biological replicates) were performed, the bars and error bars (e.g. SEM, SD) and the test used to calculate p-values in the respective figure legends. Please provide statistical testing where applicable, and also add a paragraph detailing this to the methods section. See:

<http://www.embopress.org/page/journal/14693178/authorguide#statisticalanalysis>

9) Please format the references according to our journal style. See:

10) Please add a paragraph detailing the author contributions next to the conflict of interest statement.

Finally, we require that the corresponding author supplies an ORCID ID. Please find instructions on how to link your ORCID ID to your account in our manuscript tracking system in our Author guidelines: <http://www.embopress.org/page/journal/14693178/authorguide#authorshipguidelines>

I look forward to seeing a revised version of your manuscript when it is ready. Please let me know if you have questions or comments regarding the revision.

REFeree REPORTS

Referee #1:

The study by Iavarone et al investigates the role of Cripto in macrophages during acute and chronic skeletal muscle regeneration. The results show that Cripto is expressed mainly by anti-inflammatory macrophages. Depletion of Cripto in the myeloid compartment prevents the maintenance of the anti-inflammatory phenotype of macrophages, through the inhibition of smad signaling. Skeletal muscle regeneration after one round of injury is not affected in cripto-macrophage depleted animals while muscle regeneration is altered after a second round of injury as well as in mdx muscle. Finally, the authors showed that in cripto-macrophage deficient muscle, the number of blood capillaries is decreased due to an increase of EndoMT through Smad signaling. Although the mechanisms by which Cripto acts on EndoMT is still to be investigated, this study provides a nice evidence that recovery (anti-inflammatory) macrophage prevent EndMT during tissue repair.

Comments:

- Figure 1C. At day 3, we can observe that Cripto+ cells are almost exclusively F4/80hi Ly6Cneg cells. At day 5, the F4/80 expression of cripto+ cells is not high. Does that mean that Cripto+ cells are F4/80low among all F4/80 cells (meaning that the cells have lost F4/80 expression between day 3 and day 5), or that this particular plot is not representative (in that case, please change the plot for a more representative one)?

- LysMCre-Cripto^{fl/fl}. First, LysM is more expressed as the cells differentiate. Second, the

LysMCre is more or less efficient depending on the flox locus. The analysis in SupplFig1 was made on 2-days post-injury macrophages and not on monocytes, and only provides the appearance of the excised gene. It would be better to see the amount of the WT locus in that population to evaluate the efficiency of the deletion of the gene. An incomplete deletion may partly explain why after one round of CTX the phenotype is very weak.

Referee #2:

This manuscript investigates the role Cripto plays in regulating macrophage polarity and effects on muscle regeneration and vascularization in cardiotoxin induced muscle damage and the mdx mouse model of Duchenne muscular dystrophy. Using conditional knockout mice in which Cripto is ablated from myeloid cells the authors indicate that loss of myeloid Cripto affects activation of TGFbeta/SMAD signaling and expansion of anti-inflammatory CD206+ macrophages. The authors indicate from the study loss of Cripto reduces vascular remodeling, endothelia-to-Mesenchymal transition, reduces muscle regeneration and exacerbates dystrophin deficient muscular dystrophy disease progression. The authors conclude that Cripto provides a direct functional link between macrophage populations and endothelial cells.

This is an interesting manuscript that provides a test of a hypothesis on the role Cripto plays in muscle inflammatory response and regulation of muscle regeneration, fibrosis and vascular remodeling. There are several concerns that need to be addressed:

Major comments:

1. What were the age of the mice used in the conditional knockout mouse studies? Is Cripto expression altered with age? Do satellite cells express Cripto? Can a soluble form of Cripto affect macrophage polarity?
2. How many animals were used in each study? The authors report N=3-4 throughout the study, however this is small sample size could result in statistical errors in the study. Power analysis should be performed and reported to justify the numbers used in the study.
3. Throughout the study the amount of Cripto is reported via FACS or QRT-PCR. Protein level expression of Cripto should be used to confirm loss of Cripto in conditional knockout mice and confirm Cripto changes observed in the studies (especially when RT-PCR is also used).
4. Fig 4: Cross-sectional area (CSA) was used to measure myofiber size. However, Ferrets minimal cross-sectional area is the accepted SOP of measuring cross-sectional area and should be used to avoid differences in sectioning between muscle samples.
5. Fig 5: Hydroxyproline (HOP) assay should be performed on the entire muscle to confirm Picosirius Red staining.
6. Fig 5: To conclude that Cripto affects muscle regeneration, Pax7 positive cells should also be counted within the muscle along with myogenin and embryonic myosin heavy chain.
7. Since from the results loss of Cripto does not completely ablate macrophage polarity and preclinical outcome measures reported, then it seems Cripto contributes to immune changes and may not be required. The authors should make sure this is clear in the manuscript that other factors along with Cripto play a role.
8. Cardiotoxin-induced damage is an accepted method of damaging muscle, but it may not be physiological. Exercise induced muscle damage should be used to confirm results obtained with Cardiotoxin.
9. The authors should comment on potential similarities and differences of Cripto on human vs mouse macrophages.

Minor comment:

1. There are several areas in the text with complex sentences and the need for paragraphs. The manuscript should be edited for clarity.

Referee #3:

In this manuscript, the authors examined Myeloid cell-specific Cripto gene KO phenotypes on

skeletal muscle since Cripto is expressed in macrophages. Cripto KO shows increased TGF β /Smad activation, decreased anti-inflammatory macrophages and increased endothelial cell EMT (EndMT). Eventually, muscle regeneration by CTX which was injected secondary was reduced, and muscle phenotype of mdx mice was worsened. Therefore, the authors concluded that Cripto is required for expansion of anti-inflammatory macrophages, restricted the EndMT, and proper muscle regeneration.

This manuscript is interesting to show the relationship between anti-inflammatory macrophages and EndMT via Cripto during muscle regeneration.

Major issues:

1. The authors focused on macrophages and endothelial cells as Cripto KO phenotypes since Cripto is expressed in macrophages and major phenotype of Cripto KO is increased EndMT of endothelial cells. These alteration eventually transduced reduced muscle regeneration when CTX was injected twice. However, it is not so clear how aberrant EndMT induces reduced muscle regeneration, especially during second phase of regeneration after CTX injection. It seems that reduction of satellite cell self-renewal may be the reason of the reduction of muscle regeneration after second-CTX injection. Therefore, satellite cell numbers should be quantified during first phase and second phase of muscle regeneration in both wild type and Cripto KO mice.
2. In Cripto KO, decreased capillaries after CTX injection was correlated with increased EndMT. However, it is unclear whether EndMT directly or indirectly contribute to fibrosis during muscle regeneration and mdx mice. Endothelial cell-lineage tracing analysis may be out of focus in the manuscript, but at least it should be demonstrated that EC markers/collagen co-staining in regenerating mdx muscle shows EC-derived fibrosis.

Minor issues:

1. The experimental details of "Smear plot analysis" of differentially expressed genes shown in Figure 3F are unclear.
2. The authors said the smaller regenerating fibers are increased in Cripto KO. However, Figure 4F showed that the results were opposite.
3. In Figure 8, the authors illustrated the relationship between macrophages and EMT of endothelial cells. Please integrated Cripto KO phenotypes in this illustration.

1st Revision - authors' response

23 December 2019

Reviewer #1:

The study by Iavarone et al investigates the role of Cripto in macrophages during acute and chronic skeletal muscle regeneration. The results show that Cripto is expressed mainly by anti-inflammatory macrophages. Depletion of Cripto in the myeloid compartment prevents the maintenance of the anti-inflammatory phenotype of macrophages, through the inhibition of smad signaling. Skeletal muscle regeneration after one round of injury is not affected in cripto-macrophage depleted animals while muscle regeneration is altered after a second round of injury as well as in mdx muscle. Finally, the authors showed that in cripto-macrophage deficient muscle, the number of blood capillaries is decreased due to an increase of EndoMT through Smad signaling. Although the mechanisms by which Cripto acts on EndoMT is still to be investigated, this study provides a nice evidence that recovery (anti-inflammatory) macrophage prevent EndMT during tissue repair.

We thank the Reviewer for the positive comment and for raising constructive criticisms that we have taken into account to improve the manuscript. We have now provided a revised version of the manuscript, which includes additional experiments to address his/her specific comments.

- Figure 1C. At day 3, we can observe that Cripto+ cells are almost exclusively F4/80hi Ly6Cneg cells. At day 5, the F4/80 expression of cripto+ cells is not high. Does that mean that Cripto+ cells are F4/80low among all F4/80 cells (meaning that the cells have lost F4/80 expression between day 3 and day 5), or that this particular plot is not representative (in that case, please change the plot for a more representative one)?

The Reviewer is correct that the day 5 plot was not representative. We have replaced the plot with a more representative one in the Revised Figure 1.

- LysMCre-Cripto^{fl/fl}. First, LysM is more expressed as the cells differentiate. Second, the LysMCre is more or less efficient depending on the flox locus. The analysis in SupplFig1 was made on 2-days post-injury macrophages and not on monocytes, and only provides the appearance of the excised gene. It would be better to see the amount of the WT locus in that population to evaluate the efficiency of the deletion of the gene. An incomplete deletion may partly explain why after one round of CTX the phenotype is very weak.

We thank the Reviewer for pointing out this important issue. Following the Reviewer's suggestion, the amount of WT locus was determined by genomic qPCR analysis on FACS-sorted macrophages (MPs) from Cripto^{My-LOF} and GFP-Cripto^{My-LOF} injured muscles, using sequence-specific primers to amplify the WT locus (DNA region overlapping exon 4, within the LoxP-flanked region). Sequence-specific primers were also used to amplify exon2, and used as reference PCR. The amount of Cripto WT locus was estimated over the reference PCR. DNA from heterozygous *Cripto* KO (Cripto^{+/-}) mice was used as reference of 50% of *Cripto* deletion. The estimated efficiency of *Cripto* deletion was about 50% in the overall MP population (F4/80+ MPs), and increased up to ~80% in the GFP+ monocytes/macrophage population. Data are shown in Figure EV2C and reported on page 6, line 24-28 and page 7, line 1-3. Consistently with these results, FACS analysis showed that ~50% of the MP population expressed the reporter GFP both in Cripto^{My-LOF} and Control mice. These data are shown in Figure EV2D of the revised manuscript, and reported on page 7, line 3-4.

All together these data point to an incomplete deletion of the Cripto gene, which may partly explain the weak phenotype after one round of CTX, as suggested by the Reviewer (see page 9, line 6-7) of the revised manuscript).

Referee #2:

This manuscript investigates the role Cripto plays in regulating macrophage polarity and effects on muscle regeneration and vascularization in cardiotoxin induced muscle damage and the mdx mouse model of Duchenne muscular dystrophy. Using conditional knockout mice in which Cripto is ablated from myeloid cells the authors indicate that loss of myeloid Cripto affects activation of TGFbeta/SMAD signaling and expansion of anti-inflammatory CD206+ macrophages. The authors indicate from the study loss of Cripto reduces vascular remodeling, endothelia-to-Mesenchymal transition, reduces muscle regeneration and exacerbates dystrophin deficient muscular dystrophy disease progression. The authors conclude that Cripto provides a direct functional link between macrophage populations and endothelial cells.

This is an interesting manuscript that provides a test of a hypothesis on the role Cripto plays in muscle inflammatory response and regulation of muscle regeneration, fibrosis and vascular remodeling. There are several concerns that need to be addressed:

We thank the Reviewer for the positive comment and for raising constructive criticisms that we have taken into account to improve the manuscript. In the revised manuscript, we have added new experimental data and carefully revised the text to address the specific comments and convey our message more clearly.

Major comments:

1. What were the age of the mice used in the conditional knockout mouse studies?

Mice are 10-12 weeks of age. This is now indicated on page 18, line 9 of the revised manuscript in the Materials and Methods section.

1b. Is Cripto expression altered with age?

We agree with the Reviewer that this is an interesting point to address. Indeed, we are in the progress of investigating this issue also by analyzing single cell RNA-Seq data from basal and regenerating muscle in old mice, on a collaborative basis. However, these data are not yet available and will form the basis of a future manuscript.

1c. Do satellite cells express Cripto?

Satellite cells do express Cripto after injury (*Guardiola O. et al, PNAS 2012*). This was reported in on page 4, line 10-11 of the original manuscript and the reference cited. Following the Reviewer's comment, this was further clarified in the text. See also response to point#6 below.

1d. Can a soluble form of Cripto affect macrophage polarity?

To address the reviewer's request, we have assessed the effect of a soluble form of Cripto protein (sCripto) on the MP population *in vivo*. Specifically, a biologically active purified recombinant sCripto (*Parisi, S. et al. Nodal-dependent Cripto signaling promotes cardiomyogenesis and redirects the neural fate of embryonic stem cells. J Cell Biol, 2013 163, 303-314*) was injected intramuscularly in WT mice both concomitantly with and 24 hrs after CTX -induced injury, and muscles were analysed 48hrs (day 2) after injury (see below Figure 1A). Immunofluorescence and FACS analysis showed a significant increase of F4/80+ MP population in sCripto-treated muscle compared to Control (see below Figure 1B, C top panels; and Figure 1D left panels), thus suggesting that sCripto promotes the accumulation of infiltrated MPs. These results were apparently discrepant from that observed in Cripto^{My-LOF} mice, which do not show any difference in MP accumulation (main Figure 2A, B). However, they can be explained by the non-physiological doses of sCripto and/or by the fact that different cell population may be targeted by sCripto that eventually contribute to this phenotype. Interestingly, while the percentage of Ly6C⁺/Ly6C⁻ MPs was comparable in Control and sCripto -treated mice (see below Figure 1D, right panels), the CD206+ MPs significantly increased in sCripto- treated (Figure 1B, C, bottom panels). These results fit with that observed in Cripto^{My-LOF} mice and lead to hypothesize that sCripto may promote the expansion of this MP population. However, we could not rule out the possibility that this was simply due to the increase of F4/80+ MPs in sCripto-treated muscle (Figure 1B, C, top panels, and 1D, left panels). Given the complexity of this scenario, we believe that further investigation is needed to assess the effect of sCripto on the MP populations, which would go beyond the scope of this manuscript.

These preliminary Results are not included in the manuscript and are shown in the Figure 1 below for the Reviewer.

Figure 1. Effect of soluble Cripto on macrophage populations.

(A) Schematic representation of the experimental procedure. (B) Representative pictures of immunostaining for F4/80 (green, top panels) and CD206 (red, bottom panels) in PBS- (Control) and soluble Cripto (s-Cripto)-treated TA muscle sections, at 48hrs (day2) after injury. (C) Quantification of F4/80 staining/damaged area (µm², top) and of CD206+ MPs per area (mm², bottom). Nuclei were counterstained with DAPI. Scale bar: 100 µm. Data are expressed as box plots (n=5; **P<0.01). (D) Representative flow cytometry dot plots showing the percentage of F480+ (left panels) and Ly6C⁺/Ly6C⁻ (right panels) cell populations in PBS- (Control) and soluble Cripto (s-Cripto)- treated muscles at day2 after injury. Data are mean±SEM (n=6; P=0.01).

2. How many animals were used in each study? The authors report $N=3-4$ throughout the study, however this is small sample size could result in statistical errors in the study. Power analysis should be performed and reported to justify the numbers used in the study.

We apologise with the Reviewer for the lack of clarity on this point of the original paper. The number of animals used is now clearly indicated in each experiment. Furthermore, to address the Reviewer's concern, the sample size was increased to $N \geq 5$ mice/group throughout the study. When feasible, different experimental approaches have been used to support the conclusion, as for instance immunostaining and FACS based quantification.

We would like to stress that our data are statistically significant at least at the level of a 0.05 (or below). We thus believe that there is no specific need to compute the power analysis.

3. Throughout the study the amount of Cripto is reported via FACS or QRT-PCR. Protein level expression of Cripto should be used to confirm loss of Cripto in conditional knockout mice and confirm Cripto changes observed in the studies (especially when RT-PCR is also used).

To address the Reviewer's request, Cripto protein levels were measured by ELISA assay on protein extracts from FACS sorted GFP+ Cripto^{My-LOF} and Control MPs. We could not detect Cripto protein at day 2 after injury by ELISA. This was likely due to the low abundance of the protein at this time point, which is below the detection limit of the assay. Conversely, Cripto protein levels are well detected in Control MPs at day 5 after injury and strongly decrease in Cripto^{My-LOF} MPs. These findings are in line with the FACS data showing that Cripto expression progressively increased from day 2 to 5 (Figure 1B) and confirm loss of Cripto in conditional KO mice.

These data are shown in Figure EV2, panel E and reported on page 7, line 3-4 of the revised manuscript.

4. Fig 4: Cross-sectional area (CSA) was used to measure myofiber size. However, Ferrets minimal cross-sectional area is the accepted SOP of measuring cross-sectional area and should be used to avoid differences in sectioning between muscle samples.

Following the Reviewer's request, the myofiber size analysis was repeated throughout the study using the minimal Feret's diameter (MFD), supporting our findings. These data are shown in the revised Figure 4 (panels C and F) and Figure 5 (panel J and K).

5. Fig 5: Hydroxyproline (HOP) assay should be performed on the entire muscle to confirm Picosirius Red staining.

To address the Reviewer's request, we performed the Hydroxyproline Assay on the remaining samples. Briefly, scraped 10 μ m-thick muscle sections from different parts of mdx-Control and mdx-Cripto^{My-LOF} diaphragms were processed and analyzed using the Hydroxyproline Colorimetric Assay. The amount of HOP was normalized by the volume of muscle sections to yield mg/mm³. HOP levels significantly increased in mdx-Cripto^{My-LOF} diaphragms and almost doubled compared to mdx-Control, thus confirming the Picosirius Red staining data.

Results are shown in the revised Figure 5, panel H and reported on page 10, line 19-20.

6. Fig 5: To conclude that Cripto affects muscle regeneration, Pax7 positive cells should also be counted within the muscle along with myogenin and embryonic myosin heavy chain.

To address the reviewer's comment, we performed immunofluorescence analysis with Pax7 and Myogenin (Myog) on mdx-control and mdx-Cripto^{My-LOF} diaphragms. Pax7 and eMHC antibodies are both of mouse origin and cannot be used in combination. Unfortunately, Myogenin showed a high background signal, likely due to the permeabilization step that is necessary to detect intranuclear Pax7 signals. Quantification of Myog+/Pax7- cells was thus not sufficiently reliable and has not been included. However, Pax7+ cells were quantified, and results showed no significant differences between mdx-control and mdx-Cripto^{My-LOF} mice (Figure EV4C), suggesting that the satellite cell (SC) pool was not affected, at least markedly, by myeloid Cripto ablation. These findings were also consistent with that found in the acute injury models (see response to Reviewer#3 point 1); indeed, we found only a transient decrease of Pax7+ cells in Cripto^{My-LOF} at day 5 after single injury, which was recovered at late time points (Figure EV3B, C). These findings suggest that myeloid Cripto ablation does not alter, at least markedly, the SC pool. In this context, it is important to point out that Cripto is also expressed in activated SCs after injury (*Guardiola O. et al, PNAS 2012*; and *Guardiola et al., in preparation*). Given that Cripto expression in the myogenic

compartment is not affected in *Cripto*^{My-LOF} mice, it is possible to speculate that myogenic *Cripto* has a predominant role in preserving the SC compartment. This is in line with the idea that *Cripto* exerts a cell-type (i.e. myogenic vs inflammatory) specific role in muscle regeneration (*Guardiola O. et al, PNAS 2012*). For instance, accumulation of F4/80+ MPs is not affected in SC-specific *Cripto* KO injured muscles (*Guardiola O. et al, PNAS 2012*); however, whether SC- *Cripto* KO affects MP plasticity is not known and will be the focus of future studies.

These new Results are reported on page 9, line 16-19 and page 11, line 1-3 and discussed on page 16, line 1-3 of the revised manuscript.

7. Since from the results loss of *Cripto* does not completely ablate macrophage polarity and preclinical outcome measures reported, then it seems *Cripto* contributes to immune changes and may not be required. The authors should make sure this is clear in the manuscript that other factors along with *Cripto* play a role.

Following the Reviewer's comment, we have clarified this point in the revised manuscript (see page 15 line 23-27).

8. Cardiotoxin-induced damage is an accepted method of damaging muscle, but it may not be physiological. Exercise induced muscle damage should be used to confirm results obtained with Cardiotoxin.

We respectfully disagree with the Reviewer. Intramuscular injection of Cardiotoxin is one of the most accepted procedure to induce acute muscle damage/regeneration and a widely used model of skeletal muscle regeneration. For instance, “muscle and cardiotoxin” query in PubMed retrieves more than 700 papers. This procedure allows the induction of a synchronic injury in the muscle, which allows standardizing the procedures and facilitates comparison with data from other studies. Inducing muscle damage with exercise does not permit a precise analysis of temporal parameters considering the increased in the variability of the amount of damage in terms of timing and amount. Analyzing the effect of *Cripto*^{My-LOF} in exercise induced muscle damage, although of interest, would require an entirely different experimental set up and goes beyond the scope of the present study.

9. The authors should comment on potential similarities and differences of *Cripto* on human vs mouse macrophages.

Following the Reviewer's request, a specific comment has been included on page 17, line 8-12 of the revised manuscript.

Minor comment:

1. There are several areas in the text with complex sentences and the need for paragraphs. The manuscript should be edited for clarity.

The complex sentences have been removed and the text revised to improve clarity.

Referee #3:

In this manuscript, the authors examined Myeloid cell-specific *Cripto* gene KO phenotypes on skeletal muscle since *Cripto* is expressed in macrophages. *Cripto* KO shows increased TGFb/Smad activation, decreased anti-inflammatory macrophages and increased endothelial cell EMT (EndMT). Eventually, muscle regeneration by CTX which was injected secondary was reduced, and muscle phenotype of mdx mice was worsened. Therefore, the authors concluded that *Cripto* is required for expansion of anti-inflammatory macrophages, restricted the EndMT, and proper muscle regeneration. This manuscript is interesting to show the relationship between anti-inflammatory macrophages and EndMT via *Cripto* during muscle regeneration.

We thank the Reviewer for the positive comment and for raising constructive criticisms that we have taken into account to improve the manuscript. In the revised manuscript, we have added new experimental data and carefully revised the text to address the specific comments and convey our message more clearly.

Major issues:

1. The authors focused on macrophages and endothelial cells as Cripto KO phenotypes since Cripto is expressed in macrophages and major phenotype of Cripto KO is increased EndMT of endothelial cells. These alterations eventually transduced reduced muscle regeneration when CTX was injected twice. However, it is not so clear how aberrant EndMT induces reduced muscle regeneration, especially during second phase of regeneration after CTX injection. It seems that reduction of satellite cell self-renewal may be the reason of the reduction of muscle regeneration after second-CTX injection. Therefore, satellite cell numbers should be quantified during first phase and second phase of muscle regeneration in both wild type and Cripto KO mice.

To address the Reviewer's comment, we have quantified Pax7⁺ satellite cells (SCs) both in the acute injury and in mdx models. We could detect only a transient decrease of Pax7⁺ cells in Cripto^{My-LOF} at day 5 after single injury, which was recovered at late time points (Figure EV3B and C). These findings suggest that myeloid Cripto ablation does not alter, at least markedly, the SC pool. In this context, it is important to point out that Cripto is also expressed in activated SCs after injury (*Guardiola O. et al, PNAS 2012; and Guardiola et al., in preparation*). Given that Cripto expression in the myogenic compartment is not affected in Cripto^{My-LOF} mice, it is possible to speculate that myogenic Cripto has a predominant role in preserving the SC compartment. This is in line with the idea that Cripto exerts a cell-type (i.e. myogenic vs inflammatory) specific role in muscle regeneration (*Guardiola O. et al, PNAS 2012*). For instance, we previously shown that macrophage accumulation is not affected in SC-specific Cripto KO injured muscles (*Guardiola O. et al, PNAS 2012*); yet, the impact of SC- Cripto KO on MP plasticity has to be determined and will be the focus of future studies.

These new Results are shown in Figure EV3 (panels B and C) and reported on page 9, line 16-19 and discussed on page 16, line 1-3 of the revised manuscript.

2. In Cripto KO, decreased capillaries after CTX injection was correlated with increased EndMT. However, it is unclear whether EndMT directly or indirectly contribute to fibrosis during muscle regeneration and mdx mice. Endothelial cell-lineage tracing analysis may be out of focus in the manuscript, but at least it should be demonstrated that EC markers/collagen co-staining in regenerating mdx muscle shows EC-derived fibrosis.

We would like to clarify that we do not claim a causative correlation between increased EndMT and fibrosis. We apologize for this lack of clarity. We agree with the Reviewer that endothelial cell-lineage tracing analysis would be necessary to address this issue directly; however, as also mentioned by the Reviewer, it would go beyond the scope of this manuscript.

Unfortunately, confocal analysis of diaphragm sections double stained with VE-cadherin (VEcad) and Collagen 1 (Col1) was not that informative, as Col1 signal covered most of the muscle tissue and double positive cells could not be reliably quantified. Results are shown in the Figure 2 below for the Reviewer.

Following the Reviewer's comment, and based on the above considerations the text has been rephrased (page 17, line 1-4). Furthermore, Figure 8 has been modified to convey the message more clearly.

Figure 2. Representative Confocal pictures of VEcad (green) and Col1 (red) double immunostaining on mdx-Control and mdx-Cripto^{My-LOF} diaphragm sections. Nuclei were counterstained with DAPI (blue). Scale bar: 100 μ m.

Minor issues:

1. The experimental details of "Smear plot analysis" of differentially expressed genes shown in Figure 3F are unclear.

Accepted. Additional details of smear plot analysis have been included in "Whole-genome expression analysis" paragraph of Material and methods section (page 22, line 9-12).

2. The authors said the smaller regenerating fibers are increased in Cripto KO. However, Figure 4F showed that the results were opposite.

The Reviewer is correct. The colour code inversion has been fixed in the revised Figure 4C and F.

3. In Figure 8, the authors illustrated the relationship between macrophages and EMT of endothelial cells. Please integrated Cripto KO phenotypes in this illustration.

Following the Reviewer's request, Cripto KO phenotype have been included and the Figure used as manuscript Synopsis.

2nd Editorial Decision

30 January 2020

Thank you for the submission of your revised manuscript to our editorial offices. We have now received the reports from the three referees that were asked to re-evaluate your study, you will find below. As you will see, all three referees now support the publication of the study in EMBO reports. Referee #2 has 2 remaining points I would ask you to address in a final revised manuscript. Please also provide a point-by-point response addressing these.

Further, I have these editorial requests:

- Please go through all the figure legends, including those of the Appendix, and make sure that, where applicable, the number "n" for how many independent experiments were performed and the nature of the replicates (technical versus biological) is indicated, the bars and error bars (e.g. SEM,

SD) are defined, and the test used to calculate p-values is indicated.

- Please mention the magnification boxes of the microscopic images in their legend and provide scale bars for all of these (Figs. 2A+C, 3C, 5B+I, 6A,E+F, 7A,E,H,J,K, EV3A-C, EV4C and EV5A-D). Or, indicate the magnification or the size of the boxes in the legend. Please refrain from writing the size on or near the scale bars in the image. Please add the size information only to the respective figure legend.

- Please add a legend for the Dataset EV1 as a new TAB to the respective excel file (as first TAB), providing information what the dataset contains, and what is shown in each column.

- Please upload the final Appendix file as pdf.

- Attached is the synopsis image in the size it will appear online. Some of the writing is a bit small. Could you provide this with bigger fonts, or with the small fonts in bold. Please provide this in jpeg or tiff format with the exact width of 550 pixels and a height of not more than 400 pixels.

- Finally, please find attached a word file of the manuscript text (provided by our publisher) with changes we ask you to include in your final manuscript text, and some queries, we ask you to address. Please provide your final manuscript file with track changes, in order that we can see the modifications done.

In addition I would need from you:

- a short, two-sentence summary of the manuscript
- two to three bullet points highlighting the key findings of your study

REFEREE REPORTS

Referee #1:

The authors adequately answered my questions and those of the other reviewers. This work is ready for publication. Thank you.

Referee #2:

The authors have adequately addressed most of my concerns There are a couple minor concerns that still should be addressed:

1. A small N number of animals used experimentally could result in a Type II statistical error (a false positive), so reporting Power analysis would be important along with justification from deviating from this number.
2. I agree that cardiotoxin is widely used, but it is still not considered physiological. The muscle damage caused by cardiotoxin is different from disease states and exercise-induced muscle damage. A statement indicating the potential of these differences would be acceptable.

Referee #3:

The revised manuscript has been extensively improved. I recommend acceptance.

Referee #2:

The authors have adequately addressed most of my concerns There are a couple minor concerns that still should be addressed:

We thank the Referee for the overall positive evaluation of the revised manuscript. We have addressed the remaining points by performing additional experiments/analysis and revising the text.

1. A small N number of animals used experimentally could result in a Type II statistical error (a false positive), so reporting Power analysis would be important along with justification from deviating from this number.

Following the Referee's request, we performed the Power analysis of the tests, and the results are reported in the Tables below for the Reviewer. The Power of the large majority of the tests (*t test*; $p \leq 0.05$) is $\geq 80\%$, and thus statistically powerful (*Power of a Statistical Test*; Smita Skrivanek, Principal Statistician, MoreSteam.com LLC; <https://studylib.net/doc/18300205/power-of-a-statistical-test>).

For the tests with a power lower than 80%, when feasible and compatible with a timely publication, we have increased the sample size (*n*) and the power of the tests is $\geq 89\%$ (indicated with asterisks in Table 1). These data are shown in the revised Figure EV1, and revised Figure 3. Of note, while the test of Sirius red staining quantification has a low power, the power of HOP quantification is very high ($=99\%$), thus supporting our conclusions. For the remaining tests that have a low Power, doubling the number of animals and/or repeating the bone-marrow transplantation in order to increase the power, would require a long time and is incompatible with a timely publication.

For the tests that are non-significant (*t test*; $p > 0.05$; Table 2), as expected, the Power is low and the sample size predicted is huge for most of them.

We would also like to point out that, since the main purpose underlying power analysis is to determine the smallest sample size suitable to detect the effect of a given test at the desired level of significance, it should be computed before data collection. The *post-hoc power analysis* or *retrospective power analysis* is not strictly informative; (https://www.graphpad.com/guides/prism/8/statistics/stat_why_it_isnt_helpful_to_compute.htm).

Thus, based on our data and these considerations, we believe that the statistical analysis is robust enough to support our conclusions.

Table 1. Power Analysis (*t* test; $p \leq 0.05$)

Power Analysis computed using G*Power 3.1 software						
test type: t test (two tails)						
α err prob = 0.05						
* experiments in which the "n" size has been increased						
Experiment	Figure panel	Effect size (d)	Sample size (n)	Power	Sample size (n) for Power=0.95	*
Cripto% of CD11b+ (2dpi vs 3 dpi)	1B	7,649389	5	1,00	3	
Cripto% of CD11b+ (2dpi vs 5 dpi)	1B	14,86764	5	1,00	2	
Ly6C low (day 2)	1D	5,844511	5	1,00	3	
Ly6C high (day 3)	1D	14,34445	5	1,00	2	
Ly6C low (day 3)	1D	20	5	1,00	2	
Ly6C high (day 5)	1D	13,45275	5	1,00	2	
Ly6C low (day 5)	1D	20	5	1,00	2	
qPCR WT locus	EV2C	4,509027	6	1,00	3	
qPCR Arg1 (day 5)	3B	2,867027	8	0,99	5 *	
Cripto qPCR (2d vs 5d)	EV1B	2,861662	6	0,99	5 *	
Cripto% of CD11b+ (3dpi vs 5 dpi)	1B	4,626874	5	0,99	3	
Ly6C high (day 2)	1D	4,015856	5	0,99	4	
Min Feret 5-10um (mdx)	5J	4,54528	5	0,99	3	
Capillary CSA (day 5)	6C	4,999743	5	0,99	3	
Capillary CSA (day 30)	6C	4,763361	5	0,99	3	
Capillary CSA (day 30+30)	6H	3,325715	5	0,99	4	
Capillary CSA (mdx)	6J	4,031887	5	0,99	3	
VEcad+/KLF4+ (day 5)	7B	4,593688	5	0,99	3	
VEcad+/TCF4+ (day 5)	7C	4,41156	5	0,99	3	
VEcad+/KLF4+ (day 30+5)	7L	3,948952	7 Ctrl - 8 KO	0,99	4	
qPCR Fizz1 (day 5)	3B	2,17292	8	0,98	7 *	
GFP+/CD206+ (mdx)	5D	2,84611	6 Ctrl - 5 KO	0,98	5	
VC Min Feret (mdx)	5K	3,01607	5	0,98	5	
Capillary CSA >100 (day 5)	6D	2,929656	5	0,98	5	
VEcad+/KLF4+/pSMAD3+ (day 5)	7I	3,059083	5	0,98	5	
qPCR Nos2 (day 2)	3B	2,121458	8	0,97	7 *	
CD206 density (day 5)	2E	2,491944	6	0,97	6	
HOP assay (mdx)	5H	2,851299	5	0,97	5	
CD31+/TCF4+ (day 5)	7D	2,69594	6 Ctrl - 5 KO	0,96	5	
CD206+ num (day 30+5)	EV3A	2,454447	6	0,96	6	
Capillary CSA (day 30+5)	6H	2,3145	6	0,95	6	
GFP+/CD206-/pSMAD3+ (day 5)	3D	2,500075	5	0,93	6	
Cripto qPCR (3d vs 5d)	EV1B	2,049384	6	0,89	8 *	
Cripto ELISA (day 5)	EV2E	2,27757	5	0,88	7	
GFP+/CD206+/pSMAD3- (day 5)	3D	2,152793	5	0,84	7	
VEcad+/KLF4+ (mdx)	7M	1,665129	5	0,84	11	
Pax7+ num (day 5)	EV3B	1,789056	6	0,79	10	
Capillary CSA <20 (day 5)	6D	1,855302	5	0,72	9	
Min Feret 20-25um (mdx)	5J	1,795868	5	0,7	10	
Sirius red (mdx)	5G	1,780975	5	0,69	10	
Min Feret 10-15um (30+30d)	4F	1,550379	6	0,68	12	
Min Feret 10-15um (mdx)	5J	1,734954	5	0,67	10	
eMHC num (mdx)	5M	1,711748	5	0,66	10	
CD31+/Pdgfra+ (day 5)	7F	1,613958	6 Ctrl - 5 KO	0,66	12	
Min Feret 5-7.5um (30+5d)	4F	1,63282	5	0,62	11	
Min Feret 5-7.5um (30+5d)	4F	1,569845	5	0,58	12	
Capillary num (day 5)	6B	1,529277	5	0,56	13	

Table 2. Power Analysis (*t test*; $p > 0.05$)

Power Analysis computed using G*Power 3.1 software						
test type: t test (two tails)						
α err prob = 0.05						
* experiments in which the "n" size has been increased						
Experiment	Figure panel	Effect size (d)	Sample size (n)	Power (1- β err prob)	Sample size (n) for Power=0.95	Significativity (t test)
qPCR Arg1 (day 2)	3B	0,04104421	4	0,05	15429	NO
qPCR Ltbp4 (day 2)	Appendix Figure S2	0,04734922	4	0,05	11594	NO
Pax7+ num (day 30)	EV3B	0,04777438	5	0,05	11388	NO
GFP+ (mdx)	5C	0,0489045	6 Ctrl - 5 KO	0,05	10868	NO
VEcad+/KLF4+ (day 3)	EV5D	0,06788108	6	0,05	5642	NO
F4/80% of GFP+ (day 2)	3A	0,09759	3	0,05	2730	NO
Ly6C Low% (mdx)	EV4B	0,1183629	3 Ctrl - 6 KO	0,05	1857	NO
CD11b% (3dpi vs 5 dpi)	1B	0,1270244	5	0,05	1612	NO
Ly6C High (mdx)	EV4B	0,1277823	3 Ctrl - 6 KO	0,05	1593	NO
GFP% (day 5)	3A	0,1305951	3	0,05	1525	NO
GFP+/CD206+/pSMAD3+ (day 5)	3D	0,1487188	5	0,05	1177	NO
F4/80 area analysis (day 2)	2B	0,2111806	5	0,06	584	NO
CD31-/Pdgfra+ (day 5)	7G	0,2130604	6 Ctrl - 5 KO	0,06	574	NO
Min Feret 10-12.5um (day 5)	4C	0,2686974	5	0,06	361	NO
VEcad+/KLF4+ (day 2)	EV5C	0,2864432	5	0,06	318	NO
F4/80 area analysis (day 5)	2B	0,2886631	5	0,06	313	NO
Min Feret 20-25um (day 30)	4C	0,2898098	5	0,06	311	NO
qPCR Mcp1 (day 2)	3B	0,3328377	4	0,06	236	NO
Capillary CSA (uninjured)	Appendix Figure S3C	0,3366177	5	0,07	231	NO
qPCR Mcp1 (day 5)	3B	0,3494649	4	0,07	214	NO
F4/80% (mdx)	EV4A	0,3961847	3 Ctrl - 6 KO	0,07	167	NO
CD206+/GFP+ num (day 3)	Appendix Figure S4B	0,4151343	5	0,08	152	NO
Pax7+ num (mdx)	EV4C	0,4440698	5	0,09	133	NO
CD206 density (day 2)	2D	0,503604	7 Ctrl - 6 KO	0,13	104	NO
qPCR Il4ra (day 5)	3B	0,5146874	4	0,09	100	NO
qPCR Tnfa (day 2)	3B	0,584209	4	0,01	78	NO
qPCR Ltbp4 (day 5)	Appendix Figure S2	0,6126905	4	0,11	71	NO
GFP% (mdx)	EV4B	0,6768357	3 Ctrl - 6 KO	0,13	58	NO
qPCR Nos2 (day 5)	3B	0,7974417	4	0,15	42	NO
GFP% of F4/80+ (day 2)	EV2D	0,8635764	3	0,13	36	NO
qPCR Tgfb (day 5)	3B	1,34787	4	0,16	36	NO
Capillary num (day 30+5)	6G	0,8637822	6	0,27	36	NO
Capillary num (uninjured)	Appendix Figure S3B	0,9024571	5	0,24	33	NO
Ly6C Low (day 5)	3E	0,9850817	3	0,15	28	NO
Pax7+ num (day 30+5)	EV3C	1,028136	6 Ctrl - 7 KO	0,39	26	NO
CD11b% (2dpi vs 5 dpi)	1B	1,042766	5	0,30	25	NO
qPCR Fizz1 (day 2)	3B	1,063379	4	0,24	24	NO
Capillary num (day 30+30)	6G	1,091806	5	0,33	23	NO
GFP% (day 2)	3A	1,12559	3	0,18	22	NO
Cripto qPCR (2d vs 3d)	EV1B	1,124176	6	0,42	22	NO
CD11b% (2dpi vs 3 dpi)	1B	1,152579	5	0,36	21	NO
F4/80% (day 2)	EV2D	1,241135	3	0,21	18	NO
qPCR Il10 (day 5)	3B	1,261596	4	0,32	18	NO
Pax7+ num (day 30+30)	EV3C	1,249569	5	0,41	18	NO
qPCR Tnfa (day 5)	3B	1,328317	4	0,35	16	NO
GFP+/pSMAD3+ (mdx)	5E	1,360545	6 Ctrl - 5 KO	0,51	16	NO
Capillary num (day 30)	6B	1,383359	5	0,48	15	NO
Capillary num (mdx)	6I	1,426794	5	0,5	14	NO

2. I agree that cardiotoxin is widely used, but it is still not considered physiological. The muscle damage caused by cardiotoxin is different from disease states and exercise-induced muscle damage. A statement indicating the potential of these differences would be acceptable. Following the Referee's request, a statement has been included on page 14, line 9-13 of the revised manuscript.

Accepted

10 February 2020

I am very pleased to accept your manuscript for publication in the next available issue of EMBO reports. Thank you for your contribution to our journal.

Corresponding Author Name: Gabriella Minchiotti

Journal Submitted to: EMBO reports

Manuscript Number: EMBOR-2019-49075